

# Partitioning anthropogenic and natural methane emissions in Finland during 2000–2021 by combining bottom-up and top-down estimates

Maria K. Tenkanen[1], Aki Tsuruta[1], Hugo Denier van der Gon[2], Lena Höglund-Isaksson[3],
Antti Leppänen[1,4], Tiina Markkanen[1], Ana Maria Roxana Petrescu[5], Maarit Raivonen[4], and Tuula Aalto[1]

[1]Climate Research, Finnish Meteorological Institute, 00560 Helsinki, Finland
[2]Department of Air Quality and Emissions Research, TNO, 3584 CB Utrecht, the Netherlands
[3]International Institute for Applied Systems Analysis (IIASA), 2361 Laxenburg, Austria
[4]Institute for Atmospheric and Earth System Research/Physics, Faculty of Science, University of Helsinki, 00014 Helsinki, Finland
[5]Department of Earth Sciences, Vrije Universiteit Amsterdam, 1081HV, Amsterdam, the Netherlands

**Correspondence:** Maria K. Tenkanen (maria.tenkanen@fmi.fi)

**Abstract.** Accurate national $CH_4$ emission estimates are essential for tracking progress towards climate goals. This study investigated Finnish $CH_4$ emissions from 2000–2021 using bottom-up and top-down approaches. We evaluated a global atmospheric inversion model's ability to estimate $CH_4$ emissions within a single country, focusing on how the choice of priors and uncertainties affected optimised emissions. The optimised anthropogenic and natural $CH_4$ emissions strongly depended on the prior emissions. While the range of $CH_4$ estimates was large, the optimised emissions were more constrained than the bottom-up estimates. Further analysis of CarbonTracker Europe - $CH_4$ results showed that optimisation aligned the trends of anthropogenic and natural $CH_4$ emission and improved modelled seasonal cycles of natural emissions. Comparison of atmospheric $CH_4$ observations with model results showed no clear preference between anthropogenic inventories (EDGAR v6 and CAMS-REG), but results using the largest natural prior (JSBACH-HIMMELI) best agreed with observations, suggesting that process-based models may underestimate $CH_4$ emissions from Finnish peatlands or unaccounted sources such as freshwater emissions. Additionally, using a process-model spread–based uncertainty estimate for natural $CH_4$ emissions seemed advantageous compared to the standard constant estimate. The average total posterior emission of the ensemble from one inversion model with different priors was similar to the average of the ensemble including different inversion models but similar priors. Thus, a range of priors can be used to reliably estimate $CH_4$ emissions when an ensemble of different models is unavailable.

## 1 Introduction

Methane ($CH_4$) is the second most important anthropogenic greenhouse gas (GHG) after carbon dioxide. Its atmospheric concentration has nearly tripled since pre-industrial times, largely due to human activities (Canadell et al., 2023). In recent years, especially in 2020 and 2021, the growth rate of $CH_4$ has been remarkably high (15.15 ppb in 2020 and 17.97 ppb in 2021) (Lan et al., 2024), continuing the renewed growth that began in 2007 (Nisbet et al., 2014; Mikaloff-Fletcher and



Schaefer, 2019). The reasons for this renewed growth and the record-high $CH_4$ growth rates are still under discussion (Nisbet et al., 2023), which reflects the large uncertainties in $CH_4$ emissions. Reducing $CH_4$ emissions is an effective way to mitigate climate change given $CH_4$'s short atmospheric lifetime and strong global warming potential (Forster et al., 2023; Nisbet et al., 2020; Collins et al., 2018), but to assess the success of the $CH_4$ emission reductions, we need to be able to quantify $CH_4$ emissions and their changes better than we currently do.

In the Paris Agreement, the participating countries pledged to report their GHG emissions and removals coherently and transparently by compiling national GHG inventories (NGHGI) (United Nations Framework Convention on Climate Change, 2016). The NGHGIs are evaluated jointly every five years in the global stocktake (UNFCCC) which was completed for the first time in 2023. They are based on a bottom-up approach, starting from the sources and estimating how much each source emits GHGs. The main aim is to capture trends caused by (direct) anthropogenic activities to track the effect of climate change

mitigation efforts put into practice, and thus, the NGHGIs report the emissions and sinks as annual country-totals. In addition to the NGHGIs, which each country compiles independently, there are other bottom-up anthropogenic GHG inventories composed for larger regions or even globally. Such inventories relevant for Finland are, for example, the Emissions Database for Global Atmospheric Research (EDGAR, European Commission and Joint Research Centre et al. (2023)), Copernicus Atmosphere Monitoring Service Regional inventory (CAMS-REG, (Kuenen et al., 2022)) and Greenhouse gas and Air pollution Interactions

and Synergies (GAINS, Höglund-Isaksson et al. (2020)). EDGAR is a widely used global inventory with regular updates while CAMS-REG and GAINS (the version used in this study) cover only Europe. However, both CAMS-REG and GAINS use more specific country-level data while EDGAR uses globally consistent methods. The main uncertainties in the bottom-up inventories result from the estimated magnitudes of each source category and their used emission factors. Nevertheless, they provide estimates for each source category separately.

Another way to estimate GHG emissions is to use a top-down approach, or atmospheric inversion. With a combination of atmospheric chemical transport models and atmospheric GHG mole fraction measurements, they revise the assumed prior emissions. The atmospheric inversion models of GHGs are becoming ever more important in relation to our climate politics (Leip et al., 2018). Until now, the assessment of national GHG budgets has relied on bottom-up-based inventories, especially on the NGHGIs. In the 2019 Refinement to the 2006 IPCC Guidelines for National Greenhouse Gas Inventories, the inversion

models were highlighted as a potential way to support and verify the NGHGIs (Maksyutov et al., 2019). A couple of countries (e.g. the UK (Manning et al., 2021; Lunt et al., 2021), Switzerland (Henne et al., 2016), Germany (Integrated Greenhouse Gas Monitoring System for Germany (ITMS), 2024), Australia (Luhar et al., 2020) and New Zealand (Geddes et al., 2021)) already utilise inversion modelling in their NGHGIs, either as an appendix or correcting the methods used in the NGHGI. All countries have certain advantages, for example being an island and having several atmospheric observation sites, which makes it easier

for inversion models to estimate GHGs within their national borders. However, without such advantages, partitioning inversion model results on country-level is still uncertain and shows more differences between different models and model setups (Deng et al., 2022; McGrath et al., 2023; Petrescu et al., 2023, 2024).

Atmospheric inversion models are at their strongest when estimating the total emissions, including both the anthropogenic emissions reported in NGHGIs and natural sources. The further partition to source categories is more complex, however,




but there are several methods for how this can be achieved. One way is to take advantage of prior distributions and optimise different source categories individually but simultaneously (e.g. Tsuruta et al., 2017; Segers and Nanni, 2023; Janardanan et al., 2024). With this method, since the partition between different source categories relies on the prior distribution, the optimisation is prone to miscategorise the $CH_4$ emissions if there are several sources in the same area and if the relative magnitudes of the priors are not correct. This uncertainty can be quantified to some extent by using different prior emissions and assessing how

different prior emission estimates affect the optimised emissions. Additionally, analytical inversions can be used (Cusworth et al., 2021; Worden et al., 2023). To reduce the dependence on prior distributions, usage of carbon isotopes measurements has been intensively studied (e.g. Thompson et al., 2018; McNorton et al., 2018; Basu et al., 2022; Haghnegahdar et al., 2023; Chandra et al., 2024; Mannisenaho et al., 2023). The models rely on different $CH_4$ sources having different isotope signatures, for example, emissions from wetlands have lower $\delta^{13}C$ than fossil fuel emissions. However, the sparse number of isotope

measurements limits the usage of isotope measurements in the inversions as an additional constraint. Furthermore, the isotope signatures have uncertainties (Thanwerdas et al., 2024), although, the largest uncertainty has been attributed to uncertainties in atmospheric chemistry (Basu et al., 2022).

Although only anthropogenic emissions are reported in the NGHGIs, as our climate change mitigation efforts can be targeted to them, natural GHG emissions also have an impact on climate change. Thus, it is equally important to quantify natural

emissions. In Finland, large peatland areas are a significant source of $CH_4$, and the magnitude of natural $CH_4$ emissions is high compared to anthropogenic $CH_4$ emissions, as estimated by the Finnish NGHGI (Tenkanen et al., 2023). Peatlands are concentrated in northern Finland, while the majority of the Finnish population lives in the south. Consequently, anthropogenic $CH_4$ emissions originate from the south. Different bottom-up $CH_4$ estimates, including both anthropogenic inventories and process models estimating the soil $CH_4$ balance, vary considerably in Finland. It is important to identify the reasons for these

inconsistencies. Furthermore, when interpreting the inversion model results, it is important to understand the extent to which the used prior emissions cause uncertainties in the optimised emission estimates, especially in regions where both anthropogenic and natural $CH_4$ emissions are abundant.

We studied $CH_4$ emissions in Finland during the past two decades (2000–2021) using both bottom-up and top-down approaches and discuss how the estimates from the two approaches differ. We aim to separate anthropogenic emissions from

natural peatland emissions and estimate their relative magnitudes in Finland. Our focus is on the $CH_4$ emission estimates from the inversion model CarbonTracker Europe - $CH_4$ (CTE-$CH_4$) (Tsuruta et al., 2017), used by previous studies to estimate Finnish $CH_4$ emissions (Tsuruta et al., 2019; Tenkanen et al., 2023), but here we extend the study period and investigate the results in more detail. Our study can be divided into four parts: First, we compare different anthropogenic emissions inventories (the Finnish NGHGI, GAINS, CAMS-REG and EDGAR v6, v7 and v8), both their total emission estimates and magnitudes

of different source categories. Second, we study estimates from our inversion model using different prior and uncertainty estimates and compare our ensemble with 13 inversion estimates collected in the VERIFY project (https://verify.lsce.ipsl.fr/). Third, we study the seasonal cycles of $CH_4$ emissions to see how the use of atmospheric $CH_4$ observations affects the seasonal cycle of the $CH_4$ estimates. Finally, we compare the modelled atmospheric mole fractions with observations and rank our



inversion model estimates based on this comparison, attempting to disentangle which inversion model setup agrees the best
with observations.

## 2 Materials and methods

### 2.1 Anthropogenic methane emission inventories

#### 2.1.1 Finnish NGHGI

Finnish anthropogenic $CH_4$ emissions are reported on an annual basis by Statistics Finland (Statistics Finland, 2023) following
the IPCC 2006 reporting guidelines with refinements in 2019 (Intergovernmental Panel on Climate Change, 2019). The NGHGI
Fi includes $CH_4$ emissions from agriculture, waste, energy, industry, and land use, land use change and forestry (LULUCF),
and uses a mix of Tiers 1, 2 and 3. The emissions from the fifth reporting category, (LULUCF), are not studied here because
the NGHGI Fi was the only studied inventory which reported the LULUCF emissions. However, $CH_4$ emissions from the
LULUCF sector in Finland have been discussed in detail by Tenkanen et al. (2023), and were on average 0.03 Tg yr$^{-1}$ during
2000–2021 according to the NGHGI Fi.

#### 2.1.2 EDGAR

EDGAR (https://edgar.jrc.ec.europa.eu/) is a global emission inventory developed by the Joint Research Centre of the Eu-
ropean Commission which provides estimates in a globally consistent way and does not often use country-specific details.
$CH_4$ estimates are provided by sector from agriculture, waste, energy and industry with a detailed subdivision further into
subcategories. Emissions are estimated fully based on statistical data from 1970. The latest version, EDGAR v8 (European
Commission and Joint Research Centre et al., 2023), has estimates until 2022. The spatial resolution is $0.1° \times 0.1°$ and the
temporal resolution is monthly.

#### 2.1.3 CAMS-REG

CAMS-REG-v5 is a regional European anthropogenic emission inventory for the years 2005-2018 which builds on the officially
reported emission data by the countries in the year 2020 under the LRTAP Convention and the NEC Directive for the air
pollutants and, similarly, the reported greenhouse gas emissions to UNFCCC. The structure of the dataset, harmonisation and
gap-filling approach, and proxies used to distribute the emissions spatially are described in detail by Kuenen et al. (2022). The
spatial resolution of CAMS-REG is $0.05° \times 0.1°$. The dataset provides annual total emissions by sector and is accompanied
by temporal profiles by country by sector to construct hourly emissions that can be used as model input.

#### 2.1.4 GAINS

The methodology in GAINS (Höglund-Isaksson et al., 2020) to estimate anthropogenic $CH_4$ emissions is based on the rec-
ommendation in the IPCC (2006) guidelines. For most sectors, GAINS uses country-specific information in a way that the



estimated emissions are consistent and comparable across geographic and temporal scales. More advanced methods are use to estimate emissions from solid waste sector (Gómez-Sanabria et al., 2018) and fossil and gas systems (Höglund-Isaksson, 2017). In addition to past estimates, GAINS can be used to estimate future emissions based on abatement measure scenarios. The version of GAINS used in this study includes estimates for the countries part of the European Union, Norway, the UK and Switzerland. The emissions are estimated monthly for 1990–2021 and the spatial resolution is $0.1° \times 0.1°$. The emissions from energy (upstream and downstream sources in fossil fuel extraction and use), agriculture (livestock, rice cultivation and agricultural waste burning) and waste handling (solid waste and wastewater) are estimated.

## 2.2 Atmospheric inversion model CTE-CH$_4$

The atmospheric inversion model CTE-CH$_4$ (Tsuruta et al., 2017) is based on the Bayesian inversion approach, where optimised CH$_4$ fluxes are obtained by minimising the mismatch between modelled and observed atmospheric observations, depending also on prior knowledge and uncertainties. In this study, CTE-CH$_4$ provides CH$_4$ emission estimates at a spatial resolution of $1° \times 1°$ (approximately 110 km $\times$ 40–60 km in Finland) over the northern high latitudes and at a temporal resolution of one week. CTE-CH$_4$ consists of prior flux maps representing different emission sources, the TM5 atmospheric chemistry transport model (Krol et al., 2005), and an ensemble Kalman filter data assimilation scheme (Peters et al., 2005; van der Laan-Luijkx et al., 2017). The size of the ensemble is 500 with a time lag of 5 weeks. The global horizontal resolution of TM5 is $4° \times 6°$ (latitude $\times$ longitude) but it includes a $1° \times 1°$ zoom grid over Europe with a $2° \times 3°$ zone around it. The vertical domain is divided into 25 hybrid sigma pressure levels from the surface to the upper atmosphere. ECMWF ERA5 meteorological data are used at a 3-hour resolution (Hersbach et al., 2020). Calculations include atmospheric sinks from photochemical reactions involving OH, Cl and O($^1$D). The reactions with OH is calculated based on Houweling et al. (2014). For reactions with Cl and O($^1$D), we use two schemes: using prescribed reaction rates calculated from the atmospheric chemistry general circulation model ECHAM5/MESSy1 (Jöckei et al., 2006; Kangasaho et al., 2022), and reaction rates based on Brühl and Crutzen (1993). The atmospheric sink varies from month to month but does not include interannual variability.

We use observations from a global in situ measurement network that includes the NOAA GLOBALVIEWplus ObsPack v4.0 dataset (Schuldt et al., 2021) and data from the National Institute for Environmental Studies (JR-STATION: Japan-Russia Siberian Tall Tower Inland Observation Network, Ver1.2 (Sasakawa et al., 2010)) and the Finnish Meteorological Institute (Tsuruta et al., 2019). Within Finland, measurements were collected at six sites located from southern to northern Finland, including urban, natural and marine areas (Fig. 1). Globally, our dataset included 175 stations from 2000 to 2021. Both weekly discrete air samples and hourly continuous measurements are filtered based on quality flags established by the respective institutions. To standardise the dataset, hourly continuous observations representing well-mixed atmospheric conditions were converted to daily averages, calculated from 12 to 4 pm local time, with exceptions for high mountain sites, where averages were calculated from 0 to 4 am local time, following Tsuruta et al. (2017). Observational uncertainties were quantified for each site, taking into account site-specific characteristics and measurement accuracy, and also the model's ability to predict atmospheric CH$_4$ mole fractions (Bruhwiler et al., 2014; Tsuruta et al., 2017, 2019). The uncertainties range from 4.5 to 75 ppb between global sites and from 15 to 30 ppb at the Finnish sites.





CTE-CH$_4$ optimises anthropogenic and natural fluxes simultaneously but separately at $1° × 1°$ resolution in Canada, the USA, Europe and Russia, and regionally elsewhere. The spatial correlation is defined using an exponential decay model (Peters et al., 2005), with correlation lengths of 100 km for $1° × 1°$ grid-based optimisation domains, 500 km for other land domains, and 900 km for oceanic domains.

### 2.2.1 Prior emissions

As an anthropogenic prior, EDGAR v6 (European Commission and Joint Research Centre et al., 2021) was used. Additionally, a modified version of EDGAR v6, where emissions in Europe were replaced with CAMS-REG, was used. In Finland, the CAMS-REG emissions were redistributed based on Statistics Finland's national GHG inventories of livestock and landfill distributions (see details in Tenkanen et al. (2023)). For natural prior emissions, we used estimates from two ecosystem models: JSBACH-HIMMELI (Raivonen et al., 2017; Kleinen et al., 2020) and LPX-Bern DYPTOP version 1.4 (Lienert and Joos, 2018; Stocker et al., 2014; Spahni et al., 2011, 2013), which include CH$_4$ emissions from peatlands and mineral lands. LPX-Bern DYPTOP also simulates emissions from inundated lands. In addition, natural prior from Saunois et al. (2024) was used. For other a priori sources, we used estimates from GFED v4.1s (van der Werf et al., 2017) for fire, VISIT (Ito and Inatomi, 2012) or Saunois et al. (2020) for termites, those calculated based on ECMWF data for ocean sources (Tsuruta et al., 2017) or from Weber et al. (2019), and Etiope et al. (2019) for geological emissions.

### 2.2.2 Prior uncertainty estimates

As default prior uncertainties, we used 80 % for terrestrial fluxes and 20 % for oceanic fluxes, assuming uncorrelated uncertainties, following the practice established in previous studies (e.g. Tsuruta et al., 2017; Bruhwiler et al., 2014). Since the uncertainty depends on the prior flux, this means that when the prior flux is small, the uncertainty is also small, i.e. we trust the emission more. If we not only optimise the total emissions but also try to separate different categories such as anthropogenic and natural emissions, this could lead to a misallocation, especially in regions where both anthropogenic and natural sources are prominent. The optimisation may not be able to change the correct category because the uncertainties are relatively too small, or because the uncertainties in the other category are too large (and therefore the optimisation is more likely to change those emissions). The process-model-based studies have shown that estimates of CH$_4$ emissions from wetlands vary substantially and thus have large uncertainties (Melton et al., 2013; Saunois et al., 2020; Ito et al., 2023; Chang et al., 2023). Therefore, we took advantage of this large range of different estimates of wetland methane emissions and defined the uncertainty of the natural prior emissions based on a process-model ensemble.

To calculate the uncertainties, we used an ensemble of six process-based models (Global Carbon Project, Saunois et al. (2020)). This ensemble used prognostic runs in which the models used their own internal approach to estimate the area and dynamics of wetlands. This enabled us to account for the differences in the location of wetlands, which is one of the largest uncertainties in modelling regional wetland emissions. The following models were included: LPX-Bern, JULES, ORCHIDEE, ELM, VISIT and LPJ-WSL. There was also a prognostic version of CLASS-CTEM, but we excluded it because of its coarse





**Table 1.** List of inversion setups.

| Inversion | Anthropogenic prior | Natural prior | Natural unc | Years |
|---|---|---|---|---|
| Inv$_{JSBACH\_CAMSREG}$ | CAMS-REG | JSBACH-HIMMELI | 80 % | 2004–2020 |
| Inv$_{LPX\_CAMSREG}$ | CAMS-REG | LPX-Bern DYPTOP | 80 % | 2004–2020 |
| Inv$_{LPX\_EDGAR}$ | EDGAR v.6 | LPX-Bern DYPTOP | 80 % | 2000–2021 |
| Inv$_{LPX\_EDGAR\_UNC}$ | EDGAR v.6 | LPX-Bern DYPTOP | varying | 2010–2021 |
| Inv$_{GCP\_EDGAR}$ | EDGAR v.6 | Saunois et al. (2024) | 80 % | 2000–2021 |

resolution and anomalously high values in the tropics. The average monthly values of the used process-based models in northern
and southern Finland are shown in Supplementary Fig. S1.

We made the test for the post-2010 period, so we calculated monthly averages for the 2010–2017 period. The uncertainty
was then defined monthly and also independently for each optimisation region, i.e. at the resolution of $1° \times 1°$ in the high
northern latitudes and regionally in the other regions. For the calculation of the uncertainties, we took the second and third
quartiles from the monthly estimates of the process-based model, e.g. the range of the lowest and highest 25 %. This value
was then divided by the natural prior used (LPX-Bern DYPTOP) to obtain the proportional values needed to calculate the prior
covariance matrix. The maximum uncertainty was set at 500 % and the minimum at 10 %.

### 2.2.3 Ensemble of CTE-CH$_4$ inversions

In this study, we formed an ensemble of five CTE-CH$_4$ inversion runs which had different anthropogenic and natural priors,
as well as different uncertainty estimates for the natural prior emissions. The inversion model setups are listed in Table 1.
The name of an inversion setup includes the used prior (Inv$_{natural\_anthropogenic}$). The experiment with varying natural uncertainty
estimates was done using the same priors as Inv$_{LPX\_EDGAR}$ and is noted with adding a subscript "UNC" at the end setups name
(Inv$_{LPX\_EDGAR\_UNC}$). Other differences between the setups (used priors and atmospheric sinks) are listed in Supplementary
Table S1.

### 2.3 Ensemble of VERIFY inversions

VERIFY project (https://verify.lsce.ipsl.fr/) was a Research and Innovation project funded by the European Commission under
the H2020 programme in 2018–2022. Within the project, a system to estimate GHG to support NGHGI reporting was devel-
oped. The project focused on the three major anthropogenic GHGs: carbon dioxide, methane and nitrous oxide. Here, we used
CH$_4$ inversion model results from the VERIFY project, of which some were conducted within the project and the rest of the
estimates were gathered from other projects, such as Global Carbon Project (Saunois et al., 2020).
The VERIFY ensemble consisted of 14 CH$_4$ inversion model estimates but since one of them was Inv$_{GCP\_EDGAR}$, we excluded
that from the VERIFY ensemble as it was already included in our CTE-CH$_4$ ensemble. We only investigated the total CH$_4$
emission because only 3 of the 13 VERIFY ensemble members gave the partition to natural and anthropogenic emissions. There



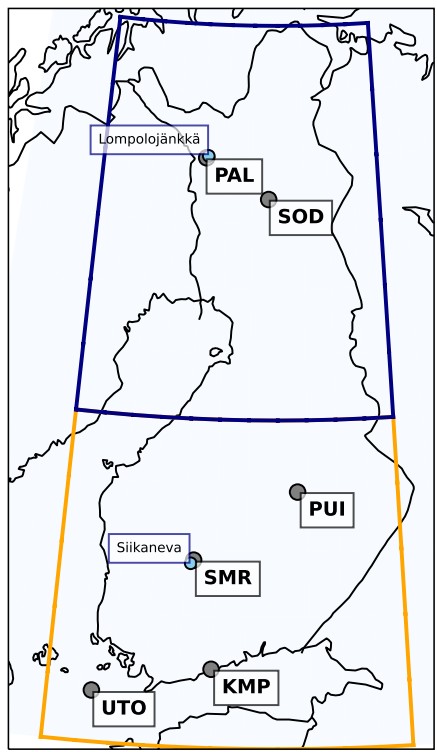

**Figure 1.** Locations of six Finnish atmospheric observation sites (black and old) and flux measurement sites (blue). The squares show the division between northern and southern Finland.

were three inversion model estimates which did not include prior emission estimates: two inversion runs with FLEXINVERT provided by the Norwegian research institute NILU, and one inversion run with FLExKF provided by the Swiss institute

EMPA.

## 2.4 Auxiliary CH$_4$ data

### 2.4.1 Eddy covariance CH$_4$ flux measurements

To verify the inversion model CH$_4$ emissions, we used eddy covariance measurements from two Finnish pristine open peatland sites, Lompolojänkkä and Siikaneva. Lompolojänkkä (68.0° N, 24.2° E) is located in northern and Siikaneva (61.8° N, 24.2° E)

in southern Finland (Fig. 1). A more detailed description of Lompolojänkkä has been given by Aurela et al. (2015) and Siikaneva by Rinne et al. (2018). Eddy covariance is an atmospheric measurement technique in which vertical turbulent fluxes are measured frequently within the atmospheric surface layer. The footprint of the measurement, i.e., where the CH$_4$ fluxes measured originate, varies depending on the prevailing meteorological conditions but is aimed to cover the whole peatland




ecosystem. The measurement were taken from the FLUXNET-CH$_4$ dataset (Delwiche et al., 2021), and the gapfilled daily
values were used here to calculate monthly averages. Lompolojänkkä had data from 2006–2010 and Siikaneva 2013–2018.

### 2.4.2 Freshwater CH$_4$ emissions

Freshwaters were defined here similar to Saunois et al. (2020), including lakes, ponds, reservoirs, streams and rivers. The
freshwater CH$_4$ emission estimates studied here were from Stavert et al. (2022), which estimated the global annual freshwater
CH$_4$ to be of 53 Tg yr$^{-1}$.

### 2.4.3 Biomass-burning CH$_4$ emissions

Two biomass-burning CH$_4$ emission estimates were used: GFED v4.1s (van der Werf et al., 2017), which was also used as
a prior in the inversions, and Copernicus Atmosphere Monitoring Service (CAMS) Global Fire Assimilation System (GFAS)
(Kaiser et al., 2012). GFED is provided in 0.25° × 0.25° and GFAS in 0.1° × 0.1° resolutions. GFED has monthly and GFAS
daily temporal resolution. We aggregated both dataset to 1° × 1° and monthly resolutions.

## 3 Results

### 3.1 Anthropogenic methane emission inventory estimates in Finland

The annual emission estimates for the four main source categories defined in the 2006 IPCC Guidelines for National Green-
house Gas Inventories (Eggleston et al., 2006) (energy, industrial processes and product use, agriculture and waste) are shown
for each inventory in Fig. 2. The spatial distribution of CAMS-REG, GAINS and EDGAR v6 can be found from Supplementary
Fig S3. Of the six inventories examined, NGHGI Fi, CAMS-REG and GAINS were in good agreement (Fig. 2a). CAMS-REG
used reported Finnish national data, and, overall, the emissions in the two inventories had similar magnitudes and trends. The
small differences between the values examined here could be because CAMS-REG has gridded estimates and the values in
Finland were obtained using an area mask, whereas NGHGI Fi is not spatially distributed. GAINS emissions were at the same
level as the other two, but there were some differences: waste emissions had a larger decreasing trend and agricultural emis-
sions were larger from 2000 to 2015. The driving cause for the trend in GAINS was declining number of cows and cattle, as
was the case also in NGHGI Fi. According to NGHGI Fi, the number of cattle decreased by more than one third between 1995
and 2021, but the decline slowed down after 2010. The decline number of cattle has also been counterbalanced by increased
animal weight, growth and milk production which has lead to larger emission factors, and thus, the magnitude agriculture CH$_4$
emissions remained the same during the latest years.

The three EDGAR versions differed from the other inventories but were similar with each other. The magnitudes of CH$_4$
emissions of different emission categories in EDGAR v6 and v7 were the same (Fig. 2b). The difference between the two
versions was that v7 had a longer time series until 2021, while v6 ended in 2018. Agricultural and industrial emissions in all
EDGAR versions were similar to those in the other inventories. Energy emissions in EDGAR v8 were smaller and showed





**Figure 2.** Annual anthropogenic $CH_4$ emissions per source category in Finland reported by a) NGHGI Fi, CAMS-REG and GAINS, and b) EDGAR v6, v7 and v8. Note the different y-axis ranges. c) Annual total $CH_4$ emissions in all the six inventories.



no trend, similar to NGHGI Fi, CAMS-REG and GAINS. However, the absolute magnitude was still three times higher in
EDGAR v8 (0.03 Tg yr$^{-1}$) than in the other inventories (0.01 Tg yr$^{-1}$). In EDGAR v6 and v7, energy emissions increased
significantly from 2004 onwards. This increase was due to increased estimates of fugitive methane emissions from oil refining
and methane emissions from natural gas processing, transmission and distribution (Supplementary Fig 2). Waste emissions,
although decreasing over the period 2000–2021, were about 4–5 times higher in the EDGAR inventories than in the other
inventories, although they were lower in v8 than in v6 and v7.

Waste emissions were the largest source in all inventories at the beginning of the study period. Due to the decreasing trend
of waste emissions, agriculture was the largest source after 2008 according GAINS and after 2009 according to NGHGI Fi. In
CAMS-REG, the emissions from waste were higher than the emissions from agriculture in 2006, otherwise the emissions from
agriculture were the highest.

The EDGAR estimates stood out when examining the total annual emissions (Fig. 2c). While the average from 2000–2020
was about 0.19 Tg yr$^{-1}$ in NGHGI Fi and GAINS, it was 0.76 Tg yr$^{-1}$ in EDGAR v7 and 0.58 Tg yr$^{-1}$ in EDGAR v8.
The EDGAR inventories also showed a larger interannual variability, especially in the waste sector, than the other inventory
estimates.

## 3.2 Atmospheric inversion model emission estimates in Finland

### 3.2.1 Annual estimates from CTE-CH$_4$

In this section, we study the annual total, anthropogenic and natural CH$_4$ emissions in Finland and how the posterior emissions
differed from the prior emissions depending on the inversion model setup. The spatial distribution of the prior and posterior
emissions can be found from Supplementary Fig. S3–S6.

Finland's annual total emission estimates are shown in Fig. 3. The prior emissions had evidently a strong influence on the
posterior emissions. The range of the total prior emissions was large, but the range of optimised emissions was smaller after
2009 and until 2020 (the average range was 0.57 Tg yr$^{-1}$ in 2009–2020) than the range of prior emissions (0.69 Tg yr$^{-1}$). The
inversions using LPX-Bern DYPTOP as the natural prior estimates had the highest (Inv$_{LPX\_EDGAR}$, 1.1 Tg yr$^{-1}$ on average) and
lowest (Inv$_{LPX\_CAMSREG}$, 0.51 Tg yr$^{-1}$ on average) total emission estimates, while the three estimates between them agreed
well, especially after 2016. To better explain the differences between the total emission estimates, the anthropogenic and natural
CH$_4$ emissions are studied separately below.

Magnitudes of the two anthropogenic posterior emission estimates using CAMS-REG were similar and slightly higher
than CAMS-REG (Fig. 4). The optimised results using EDGAR v6 varied more but all posterior emissions were higher than
EDGAR v6 until 2009 and lower thereafter until 2020 or 2021, bringing the posterior estimates of the five inversions closer
together compared to their prior estimates. The two anthropogenic posterior emissions using EDGAR v6 combined with the
default uncertainty estimate for the natural prior (Inv$_{LPX\_EDGAR}$ and Inv$_{GCP\_EDGAR}$) had similar anthropogenic emission es-
timates, but the inversion with varying uncertainty estimates for the natural prior (Inv$_{LPX\_EDGAR\_UNC}$) had lower optimised
anthropogenic emission estimates than the other two, especially after 2016.



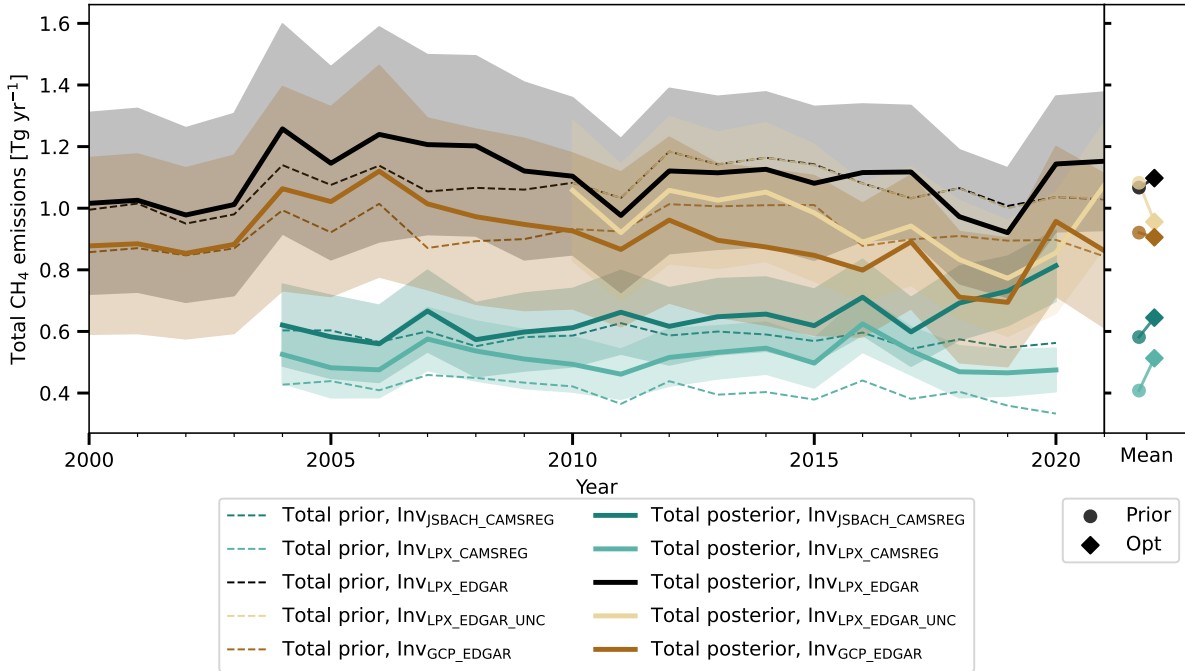

**Figure 3.** Annual total $CH_4$ emission estimates from the five CTE-$CH_4$ inversion model runs. Prior estimates are shown with dashed and posterior estimates with solid lines. Shaded areas around the posterior emissions show one standard deviation of the ensemble distributions. The right panel shows the mean prior and posterior estimates from the whole study period.

Natural posterior emissions were always higher than the prior used, regardless of the natural prior, except in 2005 and 2006 when JSBACH-HIMMELI was higher (Fig. 5). The order of emission estimates was also preserved after the optimisation: posterior emissions of $Inv_{JSBACH\_CAMSREG}$ were the highest, $Inv_{GCP\_EDGAR}$ the smallest and posterior emissions using LPX-
Bern DYPTOP as the prior, were in between the two estimates with the varying uncertainty estimates ($Inv_{LPX\_EDGAR\_UNC}$)) being the smallest estimates. Since the estimates using GCP prior did not change a lot but the optimised emissions with JSBACH-HIMMELI were increased, the range of natural posterior emissions was larger than the range of prior emissions.

Even though the absolute magnitudes of total $CH_4$ emissions and the partition between anthropogenic and natural emissions differed between inversion runs, the trends of the emission estimates were more aligned after the optimisation. All
the anthropogenic posterior emissions had decreasing trends, even though, there was no significant trend in EDGAR v6 in 2000–2021. Likewise, there were no significant trends in the natural prior emissions, but in the optimised natural emissions, there were small positive trends with $Inv_{JSBACH\_CAMSREG}$ (0.01 Tg yr$^{-1}$, p-value 0.0003) and $Inv_{LPX\_EDGAR}$ (0.005 Tg yr$^{-1}$, p-value 0.004). Decreasing anthropogenic emissions and increasing natural emissions cancelled each other out, so that the only statistically significant trends in the total emissions were noted in $Inv_{GCP\_EDGAR}$ (-0.007 Tg yr$^{-1}$, p-value 0.03) and
$Inv_{JSBACH\_CAMSREG}$ (0.009 Tg yr$^{-1}$, p-value 0.001). The signs of the trends were the opposite, reflecting the partition between





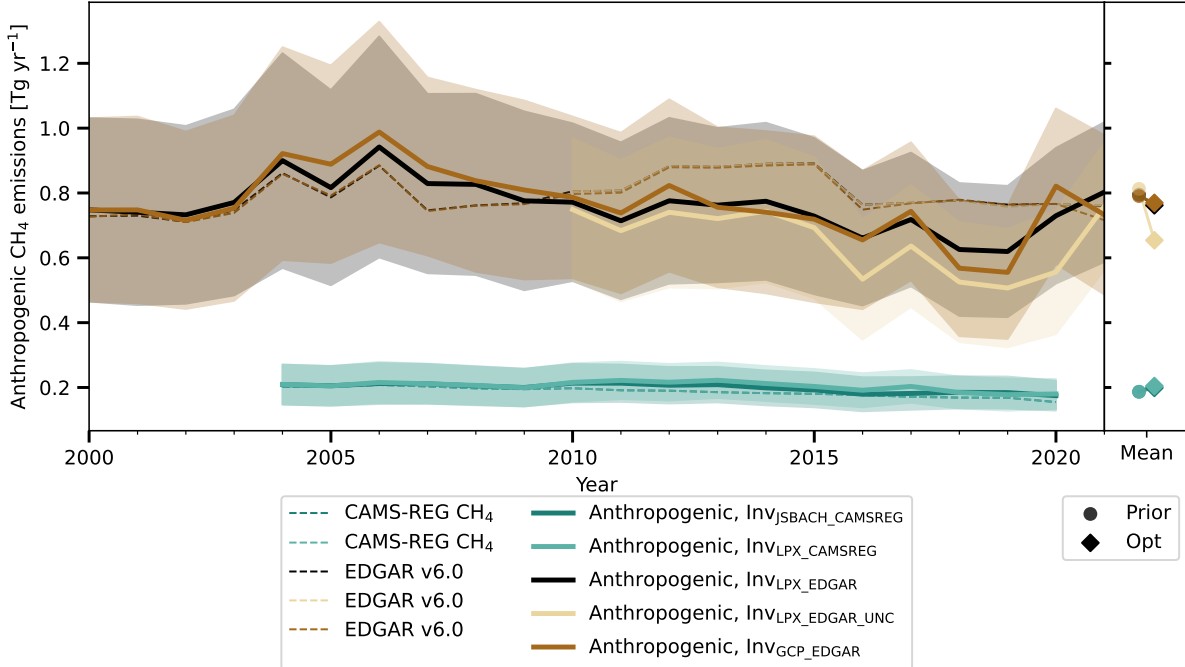

**Figure 4.** Annual anthropogenic $CH_4$ emission estimates from the five CTE-$CH_4$ inversion model runs. Prior estimates are shown with dashed and posterior estimates with solid lines. Shaded areas around the posterior emissions show one standard deviation of the ensemble distributions. The right panel shows the mean prior and posterior estimates from the whole study period.

**Table 2.** Linear trends [Gg yr$^{-1}$] and their p-values (in brackets) for anthropogenic, natural and total $CH_4$ emission estimates in Finland. Values for prior and optimised estimates from the five CTE-$CH_4$ inversion runs are shown. Statistically significant trends (p-value smaller than 0.05) are **bolded**.

| | Anthropogenic | | Natural | | Total | |
|---|---|---|---|---|---|---|
| | Prior | Optimised | Prior | Optimised | Prior | Optimised |
| Inv$_{JSBACH\_CAMSREG}$ | **-2.9 (0.00)** | **-2.2 (0.00)** | 0.8 (0.45) | **11.3 (0.00)** | -2.0 (0.07) | **9.1 (0.00)** |
| Inv$_{LPX\_CAMSREG}$ | **-2.9 (0.00)** | **-1.7 (0.01)** | -1.9 (0.16) | 1.1 (0.64) | **-4.7 (0.00)** | -0.6 (0.78) |
| Inv$_{LPX\_EDGAR}$ | 1.8 (0.36) | **-5.6 (0.03)** | 0.1 (0.87) | **4.9 (0.00)** | 2.0 (0.36) | -0.7 (0.84) |
| Inv$_{LPX\_EDGAR\_UNC}$ | **-9.6 (0.03)** | -14.5 (0.07) | 0.2 (0.92) | 1.2 (0.64) | -9.3 (0.06) | -13.2 (0.13) |
| Inv$_{GCP\_EDGAR}$ | 1.2 (0.56) | **-7.4 (0.03)** | -0.2 (0.12) | 0.3 (0.15) | 1.0 (0.63) | **-7.2 (0.03)** |



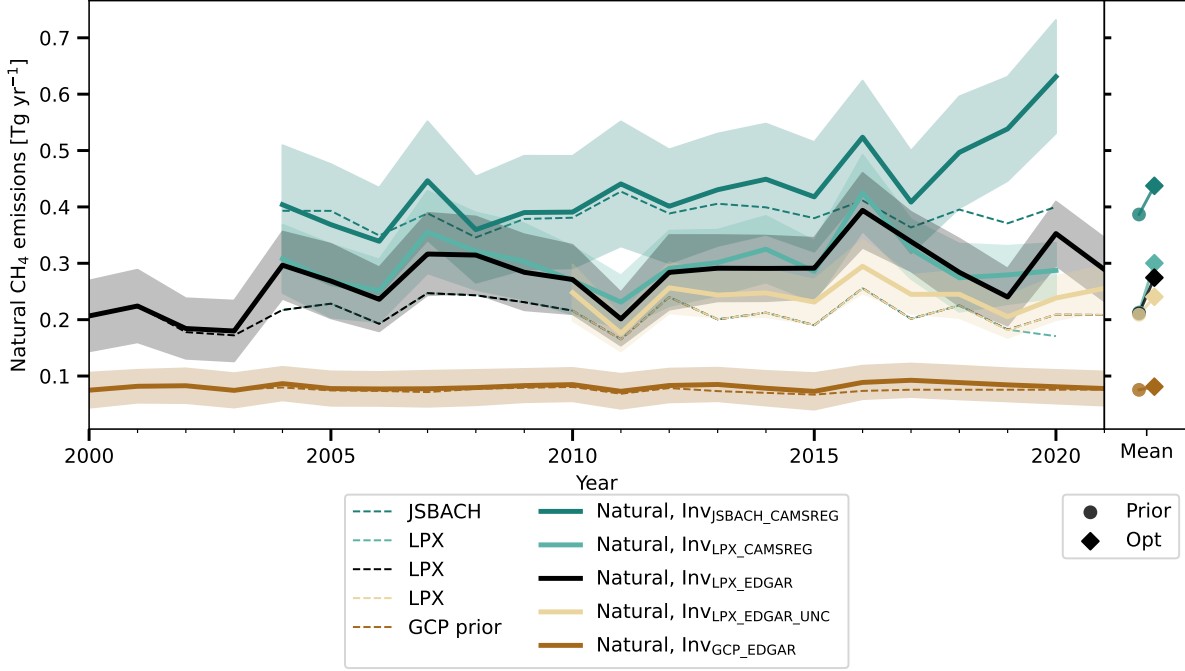

**Figure 5.** Annual natural $CH_4$ emission estimates from the five CTE-$CH_4$ inversion model runs. Prior estimates are shown with dashed and optimised estimates with solid lines. Shaded areas around the posterior emissions show one standard deviation of the ensemble distributions. The right panel shows the mean prior and optimised estimates from the whole study period.

natural and anthropogenic emissions: $Inv_{GCP\_EDGAR}$ had the highest anthropogenic emissions and the smallest natural emissions while $Inv_{JSBACH\_CAMSREG}$ had the highest natural emissions and the smallest anthropogenic emissions.

### 3.2.2 Years 2020 and 2021

During the last two years of the study period, 2020–2021, the growth rate of the global atmospheric $CH_4$ was remarkably high
(Lan et al., 2024; Nisbet et al., 2023). Although our inversion results did not show exceptionally high $CH_4$ emissions during this period, there were still some consistent signals from the inversion estimates. In 2020, all total posterior emissions were higher than in 2019. This increase was due to an increase in both anthropogenic and natural emissions, except for $Inv_{GCP\_EDGAR}$, where the increase was due to anthropogenic emissions alone. However, its natural prior and posterior emissions were small compared to the other estimates. In contrast to $Inv_{GCP\_EDGAR}$, the natural posterior emissions using JSBACH-HIMMELI as the
natural prior ($Inv_{JSBACH\_CAMSREG}$) were the largest in 2020.

In 2021 there were three posterior emission estimates, $Inv_{LPX\_EDGAR}$, $Inv_{LPX\_EDGAR\_UNC}$ and $Inv_{GCP\_EDGAR}$. All optimised total emissions were still higher than in 2019, but the differences between 2020 and 2021 diverged and the differences in natural or anthropogenic partitioning were also inconsistent. However, all estimates were higher than the prior total emissions,





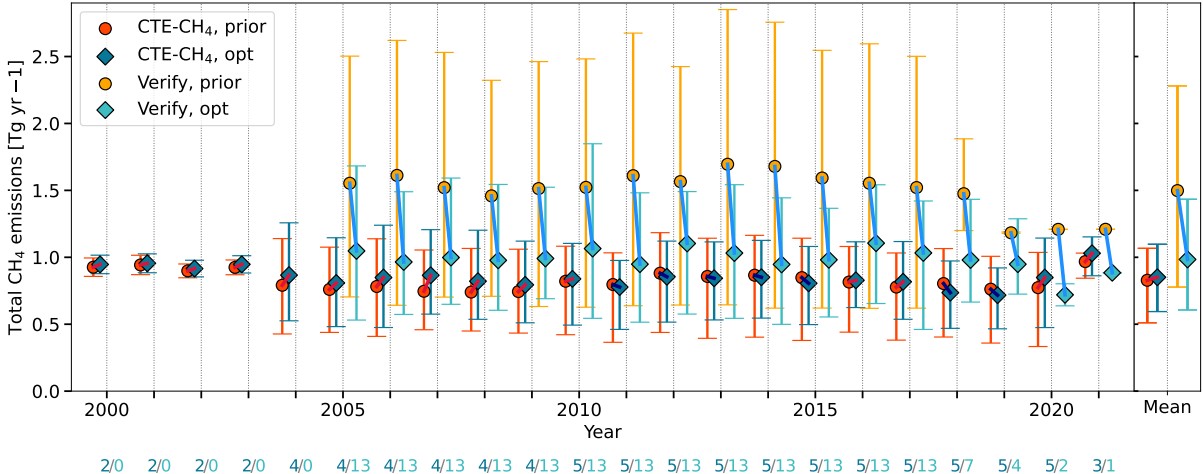

**Figure 6.** Estimated total annual $CH_4$ emissions in Finland in 2000-2021, estimated using inversion models. Two ensembles are included: CTE-CH4 and VERIFY. The circle (prior) and diamond (optimised) symbols indicate the ensemble means. The lowest and highest emission estimates are indicated by the lower and upper ends of the bars. Values for the priors are shown in yellow (VERIFY) and red (CTE-CH4) and values for the optimised estimates are shown in light blue (VERIFY) and dark blue (CTE-CH4). The mean values of the priors are connected to the mean values of the posteriors by a line. The colour of the line indicates whether the emissions have increased (red) or decreased (blue) compared to the prior. The number below the year shows the number of members of CTE-CH4 (dark blue) and VERIFY (light blue) for that year. The right panel shows the mean mean values from the whole study period.

and in particular the anthropogenic posterior estimates were close to the anthropogenic prior estimates. Part of the reason for this may be due to the high biomass-burning emission estimates in GFEDv4.1s, which seemed to affect at least the emissions in Inv$_{LPX\_EDGAR\_UNC}$. The natural posterior emission estimates of Inv$_{LPX\_EDGAR\_UNC}$ in high northern latitudes (north of 50° N) were substantially higher than the emissions from LPX-Bern DYPTOP in 2016–2020, but in 2021 the emissions decreased by 15 Tg from 2020 (Supplementary Fig. S7). In the GFED, $CH_4$ emissions in the high northern latitudes were 8.6 Tg in 2021 compared to 3.3 Tg yr$^{-1}$ in 2016–2020 (Supplementary Fig. S8). The high biomass-burning emissions in high northern latitudes most likely also affected the methane budget estimates in Finland, although there were no major forest fires in Finland. The emission estimates from another biomass-burning dataset, GFAS, were also high in 2021, but only about half of the GFED estimates (4.9 Tg, Supplementary Fig. S8).

### 3.2.3 Comparison of CTE-CH4 and VERIFY ensembles

We also compared the CTE-CH4 emission estimates to the inversion results from the VERIFY project (Fig. 6). In the CTE-CH4 ensemble, the average of the total $CH_4$ emissions from 2000–2021 was 0.83 Tg yr$^{-1}$ (average minimum and maximum range was 0.51–1.10 Tg yr$^{-1}$) in the prior and 0.85 Tg yr$^{-1}$ (0.59–1.08 Tg yr$^{-1}$) in the posterior estimates. In the VERIFY ensemble, the prior emissions were 1.50 Tg yr$^{-1}$ (0.78–2.28 Tg yr$^{-1}$), almost twice that of the CTE-CH4 ensemble, but the posterior




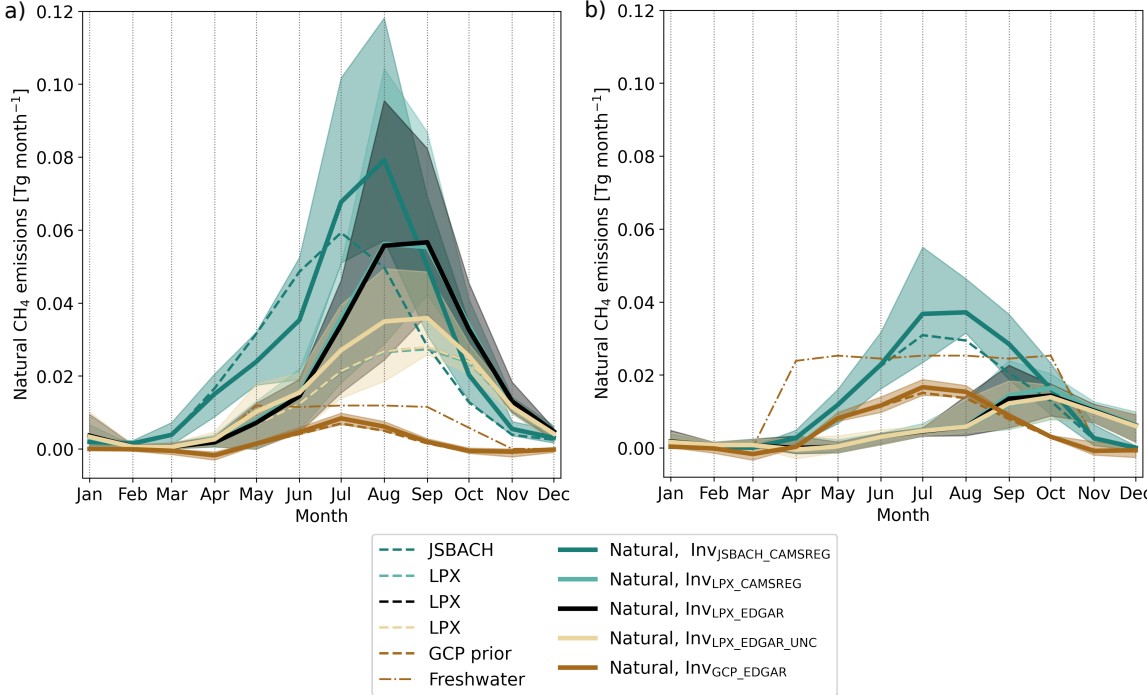

**Figure 7.** Average monthly natural CH$_4$ emission estimates from the five CTE-CH$_4$ inversion model runs in a) northern and b) southern Finland in 2010-2020. Prior estimates are shown with dashed and optimised estimates with solid lines. The shaded areas show the smallest and the largest monthly posterior emission estimates. Freshwater emissions from Stavert et al. (2022) are shown with the dash-dot lines.

emissions were reduced to 0.98 Tg yr$^{-1}$ (0.61–1.43 Tg yr$^{-1}$) bringing the total CH$_4$ emission estimates close to the CTE-CH$_4$ ensemble. The ranges of the posterior emission were large, but the range was notably smaller than the range of prior estimates,

especially with the VERIFY ensemble. The average of the posteriors in the VERIFY ensemble was close to the average of the posteriors in the CTE-CH$_4$ ensemble and within the range of the posterior CTE-CH$_4$ ensemble estimates. The lowest emission estimates from both ensembles were approximately 0.6 Tg yr$^{-1}$ but the upper limit differed by 0.3 Tg yr$^{-1}$.

### 3.3 Seasonal cycle of methane emissions

Methane emissions, and especially emissions from natural sources, have a strong seasonal cycle, and in addition to estimating
the magnitude of the emissions, it is essential to have a correct estimate of the timing of the CH$_4$ emissions. We calculated the monthly CH$_4$ emissions to see how the optimisation affected the seasonal cycle and compared the seasonal cycles of the natural emission estimates with the flux measurements from two Finnish pristine peatlands. Since the climate in southern and northern Finland is different, and thus the timing of the natural CH$_4$ emissions in southern and northern Finland is also different, we divided the emissions from 64°N according to the division used, for example, in the Finnish NGHGI (Fig. 1). We focused on
studying the emissions during the common time period between all inversion runs, 2010–2020.



The average, maximum and minimum monthly optimised natural emissions and the average of the prior emissions in 2010-2020 are shown in Fig. 7. There was a clear difference between the natural emissions in northern and southern Finland, as the maximum monthly emission estimate was almost 0.12 Tg month$^{-1}$ in the north and only half of that in the south, although both occurred in July. In addition, the timing of the maxima of the posterior emissions differed between south and north,

whereas the maxima in the priors differed only in LPX. In LPX, the maximum was either in August or September in the north and in September or October in the south, while in JSBACH-HIMMELI and the natural prior GCP the maximum was always in July. In northern Finland, the maximum of the $Inv_{GCP\_EDGAR}$ emissions did not change from July, but the maximum of the $Inv_{JSBACH\_CAMSREG}$ emissions was in August rather than in July. In southern Finland, the timing of the emissions did not change much from the priors, except in $Inv_{JSBACH\_CAMSREG}$, where the posterior emissions were shifted slightly towards late summer.

In northern Finland, posterior emissions using LPX-Bern DYPTOP ($Inv_{LPX\_CAMSREG}$, $Inv_{LPX\_EDGAR}$, $Inv_{LPX\_EDGAR\_UNC}$) had the largest increase from the prior in July–September so that the maximum was also shifted earlier towards late summer, although September was still the maximum in half of the years (Fig. 7a). This shift was less pronounced in $Inv_{LPX\_EDGAR\_UNC}$. The relative uncertainty estimates of the natural prior emissions in $Inv_{LPX\_EDGAR\_UNC}$ varied monthly and were defined independently for each $1° \times 1°$ grid cell in Finland. This meant that whether the assigned uncertainty was larger or smaller than the

constant 80 % used in the other inversions also depended on the month and location. During the winter months (November to January), the uncertainty was smaller almost everywhere in $Inv_{LPX\_EDGAR\_UNC}$. In February and March, both the natural $CH_4$ emissions and the uncertainties were small regardless of the uncertainty estimates used. From April to June, the uncertainty in $Inv_{LPX\_EDGAR\_UNC}$ was larger in northern Finland and smaller in southern Finland, but this did not have a strong effect on the posterior emissions which stayed close to the prior (Fig. 7a). From July to October, the uncertainty in the north was much

smaller in $Inv_{LPX\_EDGAR\_UNC}$ than in the other inversions, especially in grid cells where the natural prior emissions were high. Thus, the optimisation was more constrained by the prior than when the constant uncertainty was used. However, southern Finland had a larger uncertainty in summer and autumn. As the optimisation had more freedom to adjust the $CH_4$ emissions in southern Finland in $Inv_{LPX\_EDGAR\_UNC}$, it could also give more weight to the observation in the south. The natural posterior $CH_4$ emissions in the south did not differ between $Inv_{LPX\_EDGAR\_UNC}$ and $Inv_{LPX\_EDGAR}$, but the anthropogenic posterior emissions

were smaller in $Inv_{LPX\_EDGAR\_UNC}$ than in $Inv_{LPX\_EDGAR}$, especially in July–October (Supplementary Fig. S9). The smaller natural emissions in the north and anthropogenic emissions in the south led to smaller total emissions in $Inv_{LPX\_EDGAR\_UNC}$ compared to $Inv_{LPX\_EDGAR}$, and brought the seasonal cycle of total $CH_4$ emissions of $Inv_{LPX\_EDGAR\_UNC}$ close to the seasonal cycle of $Inv_{JSBACH\_CAMSREG}$ (Fig. 8a).

A comparison of the seasonal cycles of the total $CH_4$ emissions between the CTE-$CH_4$ and VERIFY ensembles (Fig. 8b)
shows similar results to the comparison of the annual totals: the VERIFY prior emissions were larger on average and had a wider range than the pior emissions in the CTE-$CH_4$ ensemble, but the amplitudes of the seasonal cycles of the average posterior emissions agreed well. However, the phases of the average posterior emissions differed: the VERIFY ensemble was consistently one month ahead of the CTE-$CH_4$ ensemble.

In Fig. 7, there are also shown the freshwater $CH_4$ emissions from Stavert et al. (2022). These emissions were not included
in the prior emissions of $Inv_{GCP\_EDGAR}$ or in the other inversions. In northern Finland, the estimated freshwater emissions were





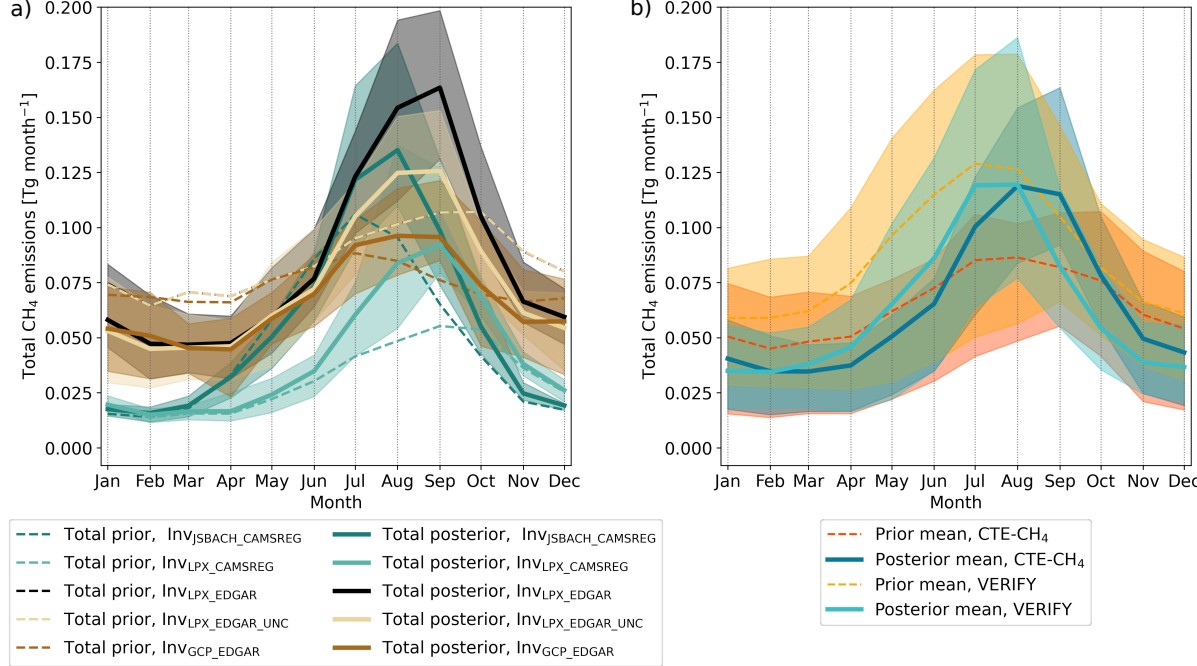

**Figure 8.** a) Average monthly total $CH_4$ emission estimates from the five CTE-$CH_4$ inversion model runs in Finland from 2010-2020. Prior estimates are shown with dashed and optimised estimates with solid lines. The shaded areas show the smallest and the largest monthly posterior emission estimates. b) Average over the monthly total $CH_4$ emission estimates from CTE-$CH_4$ and VERIFY ensembles in Finland from 2010-2020. The shaded areas show the smallest and the largest monthly estimates.

considerably smaller than the posterior natural emission estimates (except $Inv_{GCP\_EDGAR}$) but in southern Finland, where there are many shallow lakes, they were larger than the posterior natural emissions: the freshwater emissions were 0.18 Tg yr$^{-1}$ while the JSBACH-HIMMELI emissions were 0.13 Tg yr$^{-1}$ and the optimised natural emissions in $Inv_{JSBACH\_CAMSREG}$ 0.16 Tg yr$^{-1}$ on average in 2010–2020.

Figure 9 shows the measured $CH_4$ fluxes from two Finnish pristine peatlands: Lompolojänkkä (northern Finland) and Siikaneva (southern Finland). In Lompolojänkkä, the different years had very similar seasonal cycles and the maximum was in August, except in 2008 when the maximum was in July. In Siikaneva, the $CH_4$ fluxes had more annual variation. Nevertheless, the maximum of the fluxes was relatively consistent in being in July, except in 2015 and 2016 when it was in August. In both peatlands, July and August were the months with the highest $CH_4$ fluxes. Due to the alterations made by the inversion model,

the seasonal cycles of the optimised emission estimates were more consistent with the flux measurements than the seasonal cycles of the prior estimates. It should be noted, however, that the inversion model estimates aggregate much larger areas and a variety of sources rather than a single peatland, so a direct comparison between the flux measurements and inversion model results is not possible.



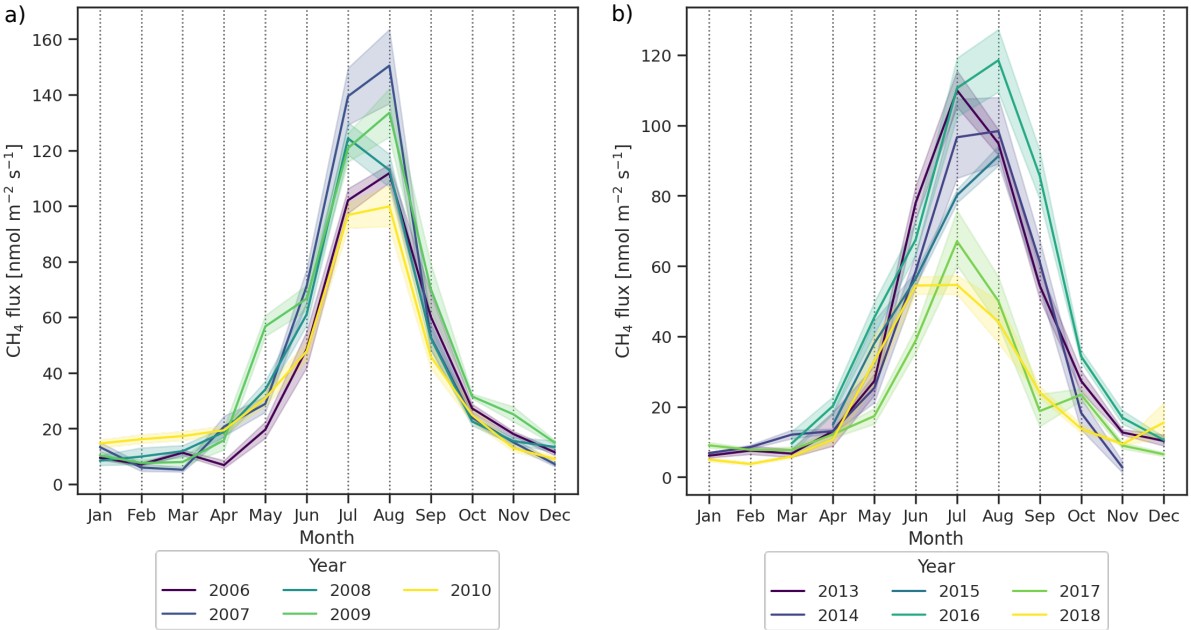

**Figure 9.** Average monthly CH$_4$ flux measurements at a) Lompolojänkkä (northern Finland) and b) Siikaneva (southern Finland). The shaded areas show a 95 % confidence interval.

## 3.4 Comparison of modelled methane mole fractions to observations in Finnish sites

The CH$_4$ emission estimates in Finland varied depending on the priors and prior uncertainty estimates used. To have an insight into which inversion best estimated CH$_4$ emissions in Finland, we compared modelled mole fractions with observed mole fractions at the six Finnish in-situ sites that were also used in the optimisation. We examined only the years 2010-2020, as these were common to all inversion runs. Observations from Utö were included from March 2018 and Hyytiälä from December 2016, since all inversions included observations from these two sites. The effect of including all available years

was also examined, but there was no significant difference. In addition to the optimised mole fractions, we also studied the mole fractions modelled with the prior emissions using a so-called forward run mode, i.e. using only the TM5 transport model without the data assimilation. The term "prior" refers to these modelled mole fractions in this section. Similarly, the term "posterior" is used to refer to the mole fractions obtained using the optimised emissions.

In Fig. 10, correlation coefficients, detrended root mean square errors (RMSE) and standard deviations are shown for all

Finnish sites as Taylor's diagrams (Taylor, 2001), comparing both the prior and posterior mole fractions. The correlation coefficient describes the linear relationship between the modelled and observed mole fractions with values close to one indicating good agreement between the two. The detrended RMSE quantifies and summarises the differences between the modelled and observed mole fractions. From the RMSE alone, it is not possible to determine whether the differences are due to different





phases in the datasets or due to differences in the amplitudes of the variations. The standard deviation provides additional
information, and it tells how much variation is in each dataset.

The statistics of the priors varied more than the statistics of the posterior mole fractions (Fig. 10, see also Supplementary
Fig. S10). The prior values of $Inv_{GCP\_EDGAR}$ were in better agreement with the observations than the other priors in almost
all observation sites, especially in Puijo, Hyytiälä and Pallas. However, in contrast to the other inversion model setups, the
posterior statistics of $Inv_{GCP\_EDGAR}$ improved only slightly from the priors at Pallas and Sodankylä, two northern stations
surrounded by natural $CH_4$ sources. Overall, the posterior mole fraction from different inversion model runs showed similar
statistics, especially at the Utö and Hyytiälä stations.

To summarise the statistics of the optimised mole fractions, we ranked selected statistics as follows: For each site and
inversion run, the bias, the detrended RMSE and the detrended linear correlation coefficient R were calculated. The bias was
used instead of the standard deviations used in Taylor's diagrams to emphasise any systematic errors in the modelled mole
fractions. Detrending the data removes long-term variations and allows us to examine short-term variations. The detrending
was done using the method introduced by Thoning and Tans (1989), which takes into account a seasonal cycle. The absolute
value of each variable was then ranked between the inversion runs from one to five, with the smallest being the best value for
the bias and the detrended RMSE and the largest being the best value for the detrended R. The average of the three rankings
for each inversion run, as well as the average of all six stations, is shown in Fig. 11. The same figure but with prior statistics is
shown in Supplementary Fig. S10.

Based on the average rankings, there was no single inversion setup that stood out as the best across all sites. Most inversion
runs had both better and worse rankings, depending on the site. However, $Inv_{LPX\_EDGAR\_UNC}$ had the best rankings in general
(average 2.11) and especially in the southern sites (Utö, Kumpula and Hyytiälä). In the northern sites (Pallas and Sodankylä),
where natural $CH_4$ sources dominate, $Inv_{JSBACH\_CAMSREG}$ had the best rankings. $Inv_{JSBACH\_CAMSREG}$ also had the second best
overall ranking (2.61). The seasonal cycles of the optimised total $CH_4$ emissions of $Inv_{LPX\_EDGAR\_UNC}$ and $Inv_{JSBACH\_CAMSREG}$
were also quite similar (Fig. 8a). $Inv_{GCP\_EDGAR}$ had the best rankings in Puijo and in Hyytiälä, where the prior statistics already
showed a good agreement with the observations, but the worst rankings in the northern peatland sites. $Inv_{LPX\_CAMSREG}$, which
had the smallest total posterior emissions, had the worst rankings in general, and $Inv_{LPX\_EDGAR}$, which had the largest total
posterior emissions, had average rankings across all sites.




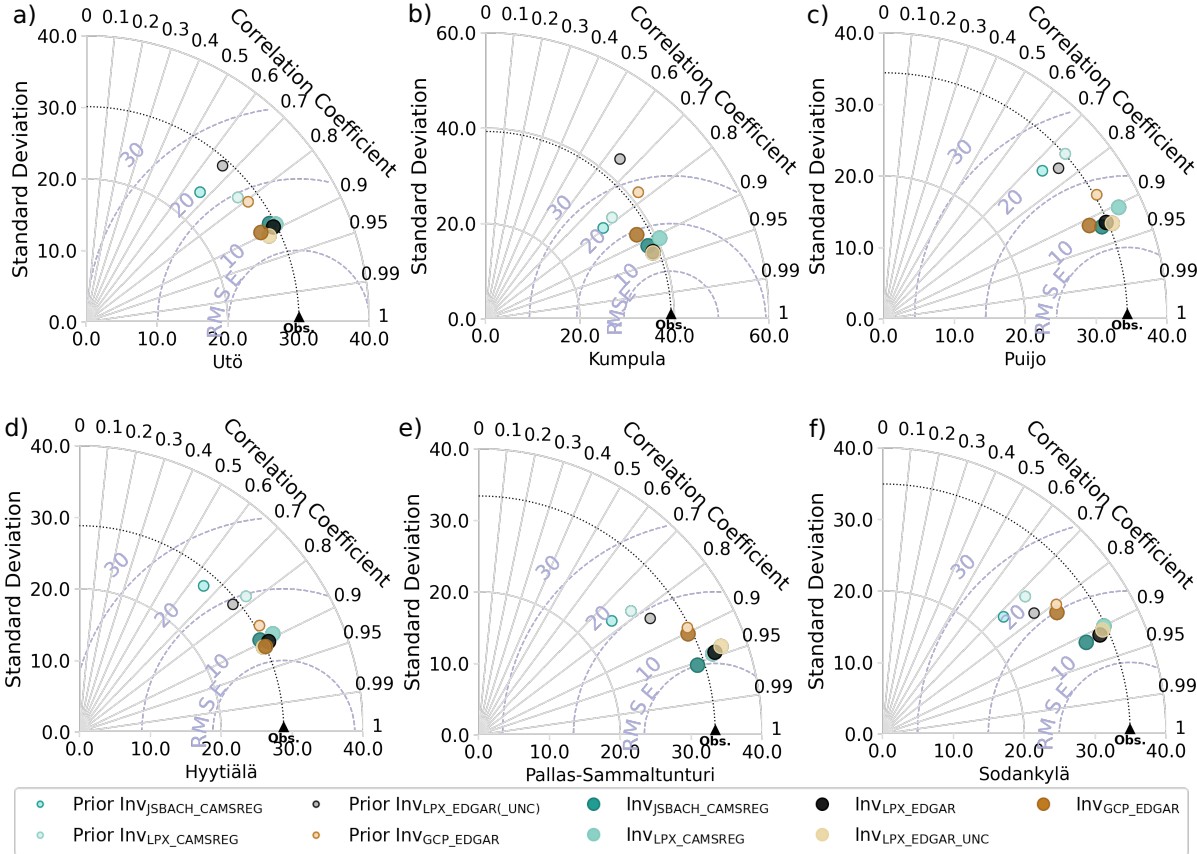

**Figure 10.** Taylor's diagram of the results of the five CTE-CH$_4$ inversion model runs against the measured mole fractions from the Finnish stations: a) Utö, b) Kumpula, c) Puijo, d) Hyytiälä, e) Pallas-Sammaltunturi and f) Sodankylä. Smaller circles respond to values from forward modelling results using the transport model TM5 and the prior emissions. Prior Inv$_{LPX\_EDGAR\_(UNC)}$ is the same for Inv$_{LPX\_EDGAR}$ and Inv$_{LPX\_EDGAR\_UNC}$ as they had the same prior emissions.

## 4 Discussion

### 4.1 Total methane emission estimates

We estimated methane emissions in Finland using the atmospheric inversion model CTE-CH$_4$. As a global model, it was able to constrain the global total emissions well (on average 525 Tg yr$^{-1}$ with a minimum and maximum range of 3.2 %). However, the ratio of the range to the average total emissions in Finland was much larger at 58 % (71 % in the priors), which shows the difficulty of constraining emissions at the country level and also how the underlying prior emissions and their distribution affect emission estimates at a smaller scale. Nonetheless, by using a global model, optimising emissions in a region of interest is not separated from emissions occurring outside the region.





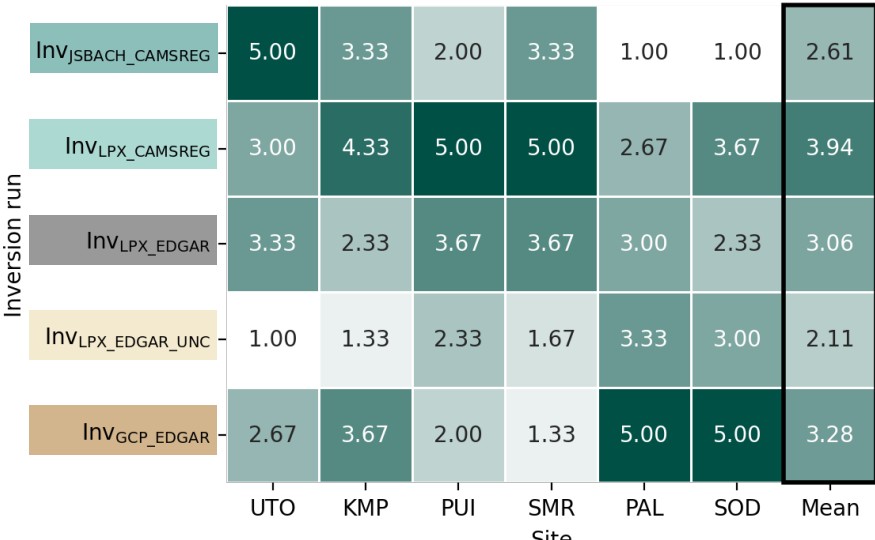

**Figure 11.** The average rank calculated for each site for each inversion run is shown. The bias, the detrended RMSE and the detrended R were calculated with each inversion run in each site and values were then ranked between the model estimates (the smallest being the best with bias and RMSE and the highest being the best with R). Additionally, the right-most column is the average over all sites averages.

The range of posterior $CH_4$ emissions in Finland was also large in the VERIFY ensemble, which included estimates from different inversion models and model runs, some of which used the same priors and observations, and some of which had

their own setups. The range of prior estimates was even wider, and since the optimisation always reduced the estimates, the largest prior estimates were most likely too high. Furthermore, the largest CTE-$CH_4$ estimate ($Inv_{LPX\_EDGAR}$), which was lower than the largest estimate in the VERIFY ensemble, and showed only moderate agreement with atmospheric $CH_4$ observations, indicating probably too high $CH_4$ emissions. Thus, the CTE-$CH_4$ ensemble range seemed to be more reliable, especially if its largest estimate were excluded. Although the ranges of posterior emissions were large, the averages of the VERIFY and

CTE-$CH_4$ ensembles agreed well. This is consistent with previous model intercomparisons which have shown that inversions can constrain emissions on a larger scale and that ensembles of inversion model estimates are more reliable and robust than estimates from a single inversion run (Petrescu et al., 2024; Saunois et al., 2020; Stavert et al., 2022). Further partitioning into independent countries still relies on the distributions of the priors.

## 4.2 Partition to anthropogenic and natural emissions

When comparing $CH_4$ estimates from the inversion model with national GHG inventories, it is important to understand not only the total $CH_4$ budget, but also the partitioning of emissions reported in the inventories. As these inventories only account for anthropogenic activities, the share of anthropogenic emissions in the total $CH_4$ estimates is particularly important. In CTE-





$CH_4$, emissions from anthropogenic and natural emissions were optimised separately but simultaneously. The emissions from both categories were analysed as they were refined by CTE-$CH_4$.

The anthropogenic emission inventories gave two drastically different estimates of Finnish $CH_4$ emissions, with EDGAR giving much higher estimates than the other inventories. Olhoff et al. (2022) compared the NGHGIs estimates with EDGAR v6 in the Nordic countries and showed that the $CH_4$ estimates from EDGAR v6 were much higher than the NGHGI estimates from Finland, Norway and Sweden. They showed that the discrepancies were due to fugitive emissions (in Norway and Finland) and waste emissions (in Sweden and Finland). Additionally, using Bayesian inversion modelling, Worden et al. (2022) estimated

Finland's waste emissions to be 0.11±0.29 Tg in 2019 instead of the prior value of 0.60±0.36 Tg (EDGAR v4.3.2 in 2012) which is much more consistent with the NGHGI Fi (0.06 Tg). Saboya et al. (2022) compared modelled mole fractions using an older version of EDGAR (v4.3.2) with observations in London and found that emissions from the waste sector were large and inconsistent with their estimates. As a global product, EDGAR uses globally consistent methodologies. As a result, there may have been a lack of consideration of country-specific mitigation strategies. For example, fugitive emissions from the oil and gas

sector in EDGAR v6 follow the trend of activity data in the Nordic countries, indicating that efforts to reduce emissions have not been taken into account (Janssens-Maenhout et al., 2019; Olhoff et al., 2022). However, in the latest update of EDGAR v8, there seems to be an improvement in the estimates for the energy sector, as they show the same trend as the other inventories in Finland (European Commission and Joint Research Centre et al., 2023).

Based on the comparison between atmospheric mole fractions modelled with CTE-$CH_4$ and observations from Finnish sites,

neither the EDGAR v6 nor the CAMS-REG seemed to be better than the other. Since the comparison with the atmospheric mole fraction does not tell directly whether the split between anthropogenic and natural emissions is correct, this could indicate that the inversion model had difficulties in separating anthropogenic and natural emissions. This is particularly likely in southern Finland, which has both anthropogenic and natural $CH_4$ sources. In addition, it can also reflect the complexity of modelling urban fluxes. To improve the estimates of anthropogenic emissions, it would be interesting to combine city-scale estimates with

our larger-scale inversions.

The three natural $CH_4$ priors used in this study differed in absolute magnitude and in spatial and temporal distribution. The comparison with the atmospheric observations from northern Finland gave a clear ranking of the three priors: $Inv_{JSBACH\_CAMSREG}$, with the highest emission estimates, seemed to have the most accurate natural estimates in Finland, inversion runs with LPX-Bern DYPTOP had the second best rankings and $Inv_{GCP\_EDGAR}$, with the lowest emission estimates, had the worst rankings.

The natural posterior emissions were always larger than their priors, even from JSBACH-HIMMELI, and the largest increases were in 2016, when the summer was warm and rainy (Finnish Meteorological Institute), and in 2019–2020. Our results indicate that Finnish natural $CH_4$ emissions from peatlands are underestimated by the process-based models, although the large natural posterior emissions could also be due to sources other than peatlands. In particular, freshwater emissions are relevant. We compared the freshwater emission estimates from Stavert et al. (2022) with the natural $CH_4$ prior and posterior emissions

in Fig. 7 and showed that in southern Finland these estimates were larger than even the highest optimised natural emissions ($Inv_{JSBACH\_CAMSREG}$). The spatial distribution of the freshwater emission estimates and the JSBACH-HIMMELI estimates coincided, so the inversion would most likely have included freshwater emissions in the posterior natural emission estimates. How-





ever, as there were still methane-emitting peatlands in southern Finland, it is not expected that optimised $Inv_{JSBACH\_CAMSREG}$ emission estimates would have included only freshwater emissions. Therefore, the freshwater emission estimates in Finland

seemed to have been too high.

## 4.3    Years 2020 and 2021

The reasons for the high atmospheric $CH_4$ growth rates in recent years, especially in 2020–2021, have been discussed. In 2020, part of the high growth rate has been attributed to a weaker OH sink caused by a decrease in $NO_x$ emissions due to Covid-19 lockdowns (Stevenson et al., 2022; Qu et al., 2022; Peng et al., 2022; Feng et al., 2023). The weaker OH sink could not explain

all the increase in atmospheric $CH_4$, and wetlands, especially at high latitudes and in the tropics, were also suggested to be responsible for the increase (Qu et al., 2022; Peng et al., 2022; Zhang et al., 2023; Feng et al., 2023). In Finland, total $CH_4$ emissions were higher in 2020 than in 2019 in all CTE-$CH_4$ inversions, and the increase was attributed to both anthropogenic and natural emissions, but the posterior natural emissions in $Inv_{JSBACH\_CAMSREG}$, which probably gave the most realistic natural emission estimates, were highest in 2020.

The increase in the $CH_4$ growth rate in 2021 has also been attributed to wetlands (Feng et al., 2023; Zhang et al., 2023). The natural $CH_4$ emission estimates from the CTE-$CH_4$ in Finland in 2021 were higher than in 2019, but at the same level or lower than in 2020, but to understand the model results, it is beneficial to study the emissions in the whole northern high latitudes. The biomass-burning product used in the CTE-$CH_4$ runs, GFEDv4.1s, estimated the $CH_4$ emissions in the boreal forests (north of 50° N) to be 8.6 Tg while they were 4.2 Tg in 2019. According to Feng et al. (2023), $CH_4$ emissions should

have been 20.8 Tg higher in 2021 than in 2019 to reproduce the observed atmospheric methane, meaning the emissions from biomass-burning in boreal forests would account for 21 % of the increase in global $CH_4$ emissions. Zheng et al. (2023) showed that $CO_2$ emissions from boreal forests have been increasing in recent decades, and that emissions were at a record high in 2021. They also used GFEDv4.1s in their analysis, but unlike our inversions, they specifically optimised emissions from biomass-burning. The record-high biomass-burning $CH_4$ emissions in the boreal forests were probably the cause for the large

decrease in the optimised wetland emissions of $Inv_{LPX\_EDGAR\_UNC}$ in the high northern latitudes from 2020 to 2021. They most likely also constrained the optimisation in Finland, keeping the posterior emissions close to the prior emissions. However, this would require further investigation and, for example, an inversion model setup where the biomass-burning $CH_4$ emissions are also optimised. There are also uncertainties in the biomass-burning emission estimates, for example GFAS had much lower emission estimates in boreal forests in 2021 (4.9 Tg, Supplementary Fig. S8). The increase in 2021 compared to 2019 was still

relatively large in GFAS (2.2 Tg), i.e. it would have explained 11 % of the global increase in $CH_4$ emissions in 2021.

## 4.4    Uncertainty estimations

In addition to different prior emissions, we also investigated how different prior uncertainties affected the emission estimates in Finland. $Inv_{LPX\_EDGAR\_UNC}$, where the natural prior uncertainty was defined based on a process model ensemble, showed better agreement with the observations at the southern sites than $Inv_{LPX\_EDGAR}$, which had the default prior uncertainties but had oth-

erwise the same setup. In northern Finland, $Inv_{LPX\_EDGAR}$ had larger uncertainties and performed better than $Inv_{LPX\_EDGAR\_UNC}$.





Additionally, $\text{Inv}_{\text{JSBACH\_CAMSREG}}$ which had the largest natural prior emissions and thus the largest uncertainties in the north, performed the best in the northern sites. One might therefore assume that large uncertainties would give the optimisation the freedom to fit the posterior emissions to the observations and with enough observations would lead to better emission estimates. However, this only seemed to be the case for natural emissions. The anthropogenic prior, EDGAR v6, had large emissions and

therefore large uncertainties, so in theory the optimisation could have reduced the anthropogenic emissions more than it did. The largest reduction from EDGAR v6 was seen with $\text{Inv}_{\text{LPX\_EDGAR\_UNC}}$, even though its anthropogenic prior uncertainty was the same as in the other runs. Thus, simply having large uncertainties and a relatively large number of sites does not guarantee a better estimate, but it is important to know where the uncertainties lie. It can be complicated to determine realistic uncertainty ranges, and even using an ensemble of multiple individual estimates might not capture the true magnitude of $CH_4$

emissions. It is good to keep in mind, though, that we used the process-based models from the lastest published GCP-$CH_4$ (Saunois et al., 2020). The ongoing effort to update the global $CH_4$ budget (Saunois et al., 2024) used updated wetland extent product (WAD2M v2.0 (Zhang et al., 2021)), and also 12 models had prognostic versions almost doubling the number of model estimates. It would be interesting to see how the uncertainty estimates would change if the process-based models from Saunois et al. (2024) were used.

The optimisation in our inversion model is based on the Ensemble Kalman Filter, which creates an ensemble of 500 members based on the priors and their uncertainties. By default, this method gives us a range of estimates that represent the uncertainties in the emission estimates. With these uncertainties we can, for example, calculate the uncertainty reduction from prior to posterior, which indicates how well the optimisation is able to constrain emissions. Another fairly robust way to estimate the uncertainties is to use different model ensembles and obtain a range of estimates (Petrescu et al., 2024; Saunois et al., 2020;

Stavert et al., 2022). As demonstrated here, using a single inversion model with different setups can constrain and produce a comparable range of $CH_4$ emission estimates at the country-level as an ensemble consisting of different inversion models. As it is easier for an individual researcher or a research group to maintain one inversion model at a time, it would be recommended to use different priors to produce more constrained and reliable $CH_4$ emission estimates.

## 5 Conclusions

This study investigated a wide range of $CH_4$ emission estimates in Finland using bottom-up (inventories and process-based models) and top-down (inversion models) approaches. We studied how the optimised emissions were affected by the setup of the inversion model runs and showed that the choice of priors strongly influences the posterior $CH_4$ emission estimates. Furthermore, our results indicate that the choice of priors affected the emission estimates as much as the choice of inversion model, and that the ensembles of inversion runs using the same inversion model but different priors resulted in similar average

total posterior emissions as when using different inversion models but similar priors.

Even though the $CH_4$ emission estimates in Finland had a large range, the range of the total posterior emissions was smaller than the range of the prior emissions. The optimisation was also able to align the trends of the anthropogenic and natural $CH_4$ emissions, and the seasonal cycles of the natural $CH_4$ emissions were altered by the optimisation so that they matched better





with the flux measurements from peatland sites. However, the spatial distributions were not radically changed from the prior
emissions.

The comparison of atmospheric CH4 observations with model results showed no clear preference between the anthropogenic inventories (EDGAR v6 and CAMS-REG), but the comparison seemed to favour the largest natural prior (JSBACH-HIMMELI). The optimised natural emissions were larger than their prior emissions, which could be due to missing emissions in the prior estimates, such as freshwater. Estimates of freshwater emissions are still highly uncertain, and the estimates ex-
amined in this study (Stavert et al., 2022) seemed to be too large for Finland. We also found evidence that emissions from biomass-burning $CH_4$, which were not optimised in CTE-$CH_4$, were likely to have a large impact on the optimised anthropogenic and natural emissions in Finland and the high northern latitudes, especially in 2021. The biomass-burning estimates used in our inversions (GFEDv4.1s) had much higher emissions than estimates from GFAS, which highlights the uncertainties in the biomass-burning $CH_4$ emission estimates.

The optimised $CH_4$ emissions were shown to depend on the choice of prior emissions. This choice was particularly important for the optimisation of the different emission components, since the optimisation of the different emission components was based on the spatial and temporal distribution of the priors. Currently there are six stations in Finland where atmospheric $CH_4$ is measured. Adding more stations would most likely help to better constrain the different emission components. In addition to more stations, we also need more reliable prior estimates and realistic uncertainty estimates. In this study, we tried a process-
model spread-based uncertainty estimate for natural $CH_4$ emissions, which appeared to be an advantageous method compared to the standard uncertainty estimates (80 % of the prior emissions). This type of uncertainty estimation could also be used for anthropogenic emissions, although, many of the anthropogenic inventories use the same statistics and activity data. However, as shown here, the choices made in compiling the inventories affect the emissions estimated, and the differences between them can help us to identify where the largest uncertainties lie.

The absolute magnitude of $CH_4$ emissions from Finland, particularly anthropogenic emissions, is relatively small compared to global totals. Consequently, these magnitudes are primarily relevant in the context of methodological comparisons or for verifying the NGHGI. The broader relevance of this study emerges from our assessment of a global model's ability to estimate $CH_4$ emissions within a single country. Such objectives are becoming increasingly relevant, as highlighted by initiatives such as the World Meteorological Organization's Global Greenhouse Gas Watch (G3W). This initiative aims to have global inver-
sion models operationally running that could be used to assess country-specific GHG emissions. Under the G3W, the inversion model results will be available in common standard formats, making them more accessible and easier to use. This will likely also encourage their use in future studies by individuals unfamiliar with inversion models. As discussed in this study, interpreting inversion model results requires careful consideration of the model setup and, in particular, posterior estimates should be considered alongside prior emissions rather than as standalone, definitive results. Ideally, those conducting the model runs
would also provide uncertainty estimates (e.g., spatial and temporal uncertainty reductions from prior to posterior, or an ensemble of inversions using different priors) and offer guidance on how to interpret the results and what factors to consider. Although preparing a comprehensive interpretation guide is challenging due to the possible diverse applications of the model results, establishing some common guidelines would be advantageous (Peters et al., 2023).



*Code and data availability.*  All the model results, inputs and code will be provided on request from the corresponding author (Maria Tenka-

nen, maria.tenkanen@fmi.fi)

*Author contributions.*  MKT, AT, and TA participated in the study design. MKT performed the data processing, prepared and performed the model runs, and prepared the visualisations for the manuscript. AT helped with the setup of the CTE-CH$_4$ and TM5 model runs and performed the model run of Inv$_{GCP\_EDGAR}$. MKT performed the analysis and preparation of the original manuscript together with TA, AT and AMRP. HDvdG provided the CMAS-REG-v5 estimates, LHI the GAINS estimates and AMRP the VERIFY inversion model estimates.

HDvdG, LHI and AMRP helped to analyse the results. AL, TM, and MR performed the JSBACH-HIMMELI model runs and provided the CH$_4$ emission estimates used in the inversion. All authors have read and approved the published version of the manuscript.

*Competing interests.*  The authors declare no conflicts of interest. Also, the funders had no role in the design of the study; in the collection, analysis, or interpretation of data; in the writing of the manuscript, or in the decision to publish the results.

*Acknowledgements.*  We thank the teams behind the LPX-Bern DYPTOP v1.4. We would also like to thank the people who worked on the

VERIFY project and the modellers who performed the inversion runs and provided the model results used in this study. We thank the European Union's project FPCUP Action 2019-2-49 "Developing supports for monitoring and reporting of GHG emissions and removals from land use, land use change and forestry" (219/SI2.818795/07 (CLIMA)) for their support. We are grateful for CSIRO Oceans and Atmosphere, Climate Science Centre (CSIRO), Environment and Climate Change Canada (EC), the Hungarian Meteorological Service (HMS), the Institute for Atmospheric Sciences and Climate (ISAC), the Institute on Atmospheric Pollution of the National Research Council (IIA),

the Institute of Environmental Physics, University of Heidelberg (IUP), Laboratoire des Sciences du Climat et de l'Environnement (LSCE), Lawrence Berkeley National Laboratory (LBNL-ARM), the Environment Division Global Environment and Marine Department Japan Meteorological Agency (JMA), the Main Geophysical Observatory (MGO), the Max Planck Institute for Biogeochemistry (MPIBGC), National Institute for Environmental Studies (NIES), Norwegian Institute for Air Research (NILU), the Pennsylvania State University (PSU), Swedish University of Agricultural Sciences (SLU) Marklund et al. (2022), the Swiss Federal Laboratories for Materials Science and Tech-

nology (EMPA), Umweltbundesamt Germany/Federal Environmental Agency (UBA), Umweltbundesamt Austria/Environment Agency Austria (EAA) as the data provider for Sonnblick, University of Bristol (UNIVBRIS), University of Exeter (Elena Kozlova), and University of Urbino (UNIURB) for performing high-quality CH$_4$ measurements at global sites and making them available through the GAW-WDCGG and personal communications. We also thank ICOS PIs for providing the data/facilities on the atmospheric CH$_4$ concentration products. In situ observations collected over the US Southern Great Plains were supported by the Office of Biological and Environmental Research of the

US Department of Energy under contract no. DE-AC02-05CH11231 as part of the Atmospheric Radiation Measurement (ARM) Program, ARM Aerial Facility (AAF), and Terrestrial Ecosystem Science (TES) Program. Measurements at Jungfraujoch were supported by ICOS Switzerland. The following AI tools were used to revise and proofread parts of the text: Grammarly (https://grammarly.com/), DeepL Write (https://www.deepl.com/en/write) and Microsoft's Copilot. Copilot was also used to assist in the plotting of the figures. Taylor's diagram in Fig. 10 were plotted using code provided by Rochford (2016).



*Financial support.* We thank the EU-H2020 VERIFY (776810), EU-Horizon EYE-CLIMA (101081395), Academy of Finland Center of Excellence (272041), FIRI - ICOS Finland (345531), and GHGSUPER (351311), Flagships ACCC and FAME (337552 and 358944), CSC (FICOCOSS) and JTF-VISIO for financial support.





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
