# Peer review of "Partitioning anthropogenic and natural methane emissions in Finland during 2000–2021 by combining bottom-up and top-down estimates"

_EGUsphere, 2024_

## Author Comment (AC1)

Replies to Reviewer 1

The following contains the comments from the reviewer in *italic* and replies by the authors in normal font.

*The authors present a study on estimating methane emissions in Finland from both bottom-up and top-down approaches. In particular, the focus lies on the comparison of an ensemble of different bottom-up estimates and, subsequently, optimizing both anthropogenic and natural $CH_4$ emissions simultaneously using different set-ups of prior estimates and uncertainties. This study shows a relevant example of estimating $CH_4$ emissions from different sources at country scale. Additionally, since the study encompasses a study period of more than 20 years, it gives a valuable insight in long-term $CH_4$ emission trends in a northern European country.*

*In my opinion, this study is well prepared and carefully reflected upon. However, I see a major weakness in the presented manuscript, namely the description and documentation.*

We thank the reviewer for the nice overview and kind comments. We answer their concerns and detailed comments below.

*I can see that the authors have put a lot of thought and effort into this study and have come to valuable and conclusive results, but the current presentation does not do justice to the work. I would strongly recommend revising the manuscript keeping in mind the following advices, which will help the reader to follow the presented work and understand the underlying train of thoughts:*

- *Provide reasons and constraints for the choices of the set-up*
- *Provide equations to explain the calculations*

- *When describing figures, start with the general findings and end with more detailed ones (e.g. first describe the whole period of time and then individual years)*
- *Be more generous with putting cross-references to other sections of the paper, where details of a topic have been discussed or will be discussed, so the reader knows where to find the corresponding information*

We thank the reviewer for these general and helpful comments. We have now revised the manuscript and tried to improve its structure and readability. We hope that it meets the reviewer's wishes. For example, we have revised Section 2.2, which describes the inverse model CarbonTracker Europe - CH4 and the choices made in the setups. We have also added equations for the cost function minimised by the inverse model (Section 2.2, line 133) and equations explaining how we defined the new uncertainty estimates of the natural prior CH4 emissions (Section 2.2.5, lines 225-233). We have added cross-references to sections and figures where appropriate.

*More specifically, in my view, there are four relevant points which should be addressed in the revision of the manuscript:*

1. *I would strongly suggest to extend Section 2.2 ("Atmospheric inversion model CTE-CH$_4$") where the details of the inverse modeling set-up is explained. I assume this paragraph was kept short since the CTE-CH$_4$ model has been explained in detail in in a different paper, but nevertheless crucial information are missing.*

   *First of all, the equation of the ensemble Kalman filter that was used in the study should be included and the different components explained. Otherwise, explanations like "The size of the ensemble is 500 with a time lag of 5 week" are incomprehensible. Additionally, some basic information on the TM5 atmospheric chemistry model (e.g. is it Eulerian or Lagrangian, is it used in forward or backward mode, was a spin-up period or additional input-data for the initial conditions used etc.) should be given. The model and inversion domain is also not very clear: first it is written that optimized CH$_4$ estimates*

*are provided over "high northern latitudes". Later it is written that the fluxes are also optimized at 1°x1° in Canada, the USA, Europe and Russia. So not only in high northern latitudes or just in the northern parts of these countries? Also, the transport model has a global resolution of 4°x6° with a zoomed grid over Europe at 1°x1° with a 2°x3° around it, but how large is this "around zone" and what area (longitude and latitude) is defined as Europe? I think it would be very helpful to illustrate this by having a map showing the different zones of the transport model and/or the inverted domain. For the observations in Finland, the author's should provide a table with the sites including their full name, exact locations and altitudes, either in the main text or the supplements, as an addition to Figure 1.*

*Moreover, I think the authors should consider splitting this section (2.2) in three different subsections: one for the transport model, one for the inversion set-up and one for the observations. If the authors prefer to keep the inversion setup and transport model together, that would also be appropriate, but in particular the section on observations (P5, L140-151) should, in my opinion, be a separate subsection.*

We thank the reviewer for their thorough comments. Indeed, we tried to keep this section relatively short because, as the reviewer points out, the model has already been published and used in several papers. However, we also agree with the reviewer that a more comprehensive description of the model would be good, especially given the scope of the journal.

We have now rewritten Section 2.2 and divided it into sections as suggested by the reviewer. We have also added a supplementary figure showing the locations of the observation sites globally as well as the optimisation regions (Fig. S1), and included tables with details of each site used (Table 1 for Finland and supplementary Table S1 for all sites).

2. *The authors have clearly put a lot of thought into estimating the prior uncertainties, which is excellent, but the explanation in Section 2.2.2 is sometimes difficult to follow.*

*First of all, which uncertainties are used for the anthropogenic fluxes, is it also 80% and 20%? Also, the calculation that are described from P6, L179 is only used for one of the inversion set-ups, which should already be highlighted in the description, otherwise it is unclear why there is a default uncertainty in the first place. It is written: "We made the test for the post-2010 period, so we calculated monthly averages for the 2010–2017 period" – which test and why only until 2017 and not 2021? I would highly recommend explaining this calculation with the help of an equation, it will be much easier to follow your description.*

We have heavily edited Section 2.2.2 (Section 2.2.5 in the revised manuscript) and added equations that hopefully help the reader to understand how the new uncertainties have been defined. We have also clarified the sentences about the default uncertainties and the period over which the inverse model was run:

> "As default prior uncertainties for both anthropogenic and natural emissions, we use 80 % for terrestrial fluxes and 20 % for oceanic fluxes, assuming uncorrelated uncertainties, following the practice established in previous studies."
> "The inverse model run with these new uncertainty estimates extends from 2010 to 2021, but the process-based models only have estimates up to 2017. Thus, we calculate monthly averages from the process-based model estimates for the period 2010—2017."

3. *It would be interesting if the authors could provide some additional information on how the area of Finland is "defined". In Figure 1 on page 8, defined areas of northern and southern Finland corresponding to Figure S1 and Figure 7 are shown. However, these areas exclude small parts of Finland and include larger parts of Sweden and Russia. Later in Section 3, the prior and posterior $CH_4$ emissions in Finland are shown. Since the optimization was at 1°x1°, these estimates probably also include emissions outside the Finish borders, which is evident from the Figures S3-S6. I think it would be nice to add*

*a clarification of the area and/or a small discussion on the share of emissions, that actually lie outside of Finland.*

Again, we agree with the reviewer and admit that the map of Finland in Fig. 1 was a bit misleading. Its aim was to show the division between northern and southern Finland, but as that was done more schematically, it is misleading. We have now modified to figure to show actually the area that was used, and also show the percentages of the grid cell along the boarded that were used to calculate the emissions in Finland.

[Figure]

4. *Throughout the paper I noticed numerous grammatical errors. I have pointed out some of these in the technical comments, but to correct them all would be beyond the scope of this review. The authors should also pay some attention*

*to tenses: sometimes the authors switch between past and present tense for no apparent reason. The grammar of the manuscript should be checked thoroughly before resubmitting the paper (I would suggest using a grammar checker for the whole text).*

We have now checked the grammar.

**Specific comments**

*P1, L19:*

*For people studying $CH_4$ it may be obvious what is meant by "renewed growth rate in 2007". However, I think it would be beneficial if you could add, for example, "renewed growth rate in 2007, after a significant decline in growth at the beginning of the millennium" for further specification.*

We have now rewritten the sentence:

> "Since atmospheric CH 4 measurements began in the late 1970s (Rice et al., 2016), the growth rate of atmospheric $CH_4$ has varied considerably, with periods of rapid growth as well as a plateau (Nisbet et al., 2023): The growth rate of $CH_4$ was close to zero from 2000 to 2006, after which the atmospheric $CH_4$ levels began to rise again (Nisbet et al., 2014; Mikaloff-Fletcher and Schaefer, 2019), reaching remarkably high increases of 15.15 ppb in 2020 and 17.97 ppb in 2021 (Lan et al., 2024).

*P2, L20:*

*When you say "record growth rates", are you referring only to 2020 and 2021? That is a bit unclear. You could put the years in brackets for clarification.*

Yes, we do. The years are now mentioned:

> "The reasons for this renewed growth and the record-high $CH_4$ growth rates in 2020 and 2021 are still under discussion (Nisbet et al., 2023), which reflects the large uncertainties in $CH_4$ emissions."

*P2, L22:*

*Please add the atmospheric lifetime and GWP of CH$_4$ in numbers for completeness.*

Added:

> "Reducing CH$_4$ emissions is an effective way to mitigate climate change (Nisbet et al., 2020; Collins et al., 2018) given the short atmospheric lifetime of CH$_4$ (9.1 years; Canadell et al., 2023) and its high global warming potential (82.5 times higher than CO$_2$ on a 20-year time scale; Forster et al., 2023)."

*P2, L27:*

*The "(UNFCCC)" is a bit confusing. Is it for clarification of the abbreviation and if so, why at this place? Or do you want to say "by the UNFCCC"?*

"UNFCCC" was referencing to a citation. It has now been modified, and the citation reads now as "UNFCCC, 2023".

*P3, L67:*

*Another strategy for constraining the sources in inverse modeling approaches is to use co-emitted gases such as Ethane, e.g. Thompson et al., 2018; Rice et al., 2016 (see references). It is not necessary to mention it, but you can consider including it for completeness.*

We agree with Reviewer and have added a mention of using co-emitted species as a constraint in an inversion model:

> "Similar to using CH$_4$ isotope measurements as an additional constraint, we can also use co-emitted species and the ratio of them to CH$_4$ emitted from specific sources, such as ethane (Rice et al., 2016; Ramsden et al., 2022; Thompson et al., 2018)."

*P3, L74:*

*You mention that the CH₄ estimates vary considerably in Finland. It would be nice if you could already include a range of annual CH₄ emissions in numbers to show how large the the discrepancies are with the corresponding sources. Alternatively, you could make a cross-reference to section 3.1, where the issue is discussed.*

Modified as suggested:

> "Different bottom-up estimates of CH₄, including both anthropogenic inventories (0.19–0.76 Tg yr−1; Section 3.1) and process models estimating the soil CH₄ balance (0.08–0.39 Tg yr−1; Section 3.2), vary considerably in Finland."

*P4, L95-L96:*

*I think it would be good to add the explanation for the abbreviation "NGHGI Fi". Even though it may be obvious that you mean the NGHGI of Finland, it is the first time you use it in this abbreviation and should therefore be explained.*

Modified as suggested.

*P4, L97:*

*Would it be possible to add an explanation of "Tiers 1, 2 and 3"? Personally, I'm not familiar with this term and I don't think it is common knowledge.*

We added short explanations:

> "The Finnish NGHGI...uses a mix of Tiers 1 (emissions factors from IPCC reports), 2 (country-specific emissions factors) and 3 (more advanced methods like process-based modelling)."

*P4: L109-L111:*

*The first sentence of this paragraph is confusing. Was CAMS-REG created in 2020? Or are the emissions from 2005-2018 based on the numbers from 2020? Or are the reports from 2020 used to create the emissions from 2005-2018 but based on the*

*corresponding years? Please reformulate this sentence and if possible, also explain the abbreviation "LRTAP" and "NEC".*

We tried to clarify this:

> "CAMS-REG v5 is a European anthropogenic emission inventory covering the period from 2005 to 2018. It builds on the emission data reported officially in 2020 by countries under the convention on long-range transboundary air pollution (UNECE, 2012) and the EU national emission ceilings directive (European Commission, 2016) for the air pollutants and, similarly, the reported GHG emissions by the countries to UNFCCC."

*P5, L144:*

*I would suggest including a map and/or a table with the 175 global stations in the supplements for completeness.*

We added it to the supplement (Fig. S1).

*P5, L148-150:*

*Could you provide one or two examples of which "site-specific characteristics" were taken into account and how they influenced the error estimation?*

We have now expanded the explanation:

> "Observational uncertainties, also referred to as "model–data mismatches", are quantified for each site by considering site-specific characteristics and measurement accuracy, and the ability of TM5 to simulate atmospheric $CH_4$ mole fractions (Bruhwiler et al., 2014; Tsuruta et al., 2017, 2019). Discrepancies between modelled and observed mole fractions are expected due to the resolution of TM5 and transport errors. For example, TM5 performs better in simulating mole fractions from remote marine background sites compared to sites influenced by strong local emissions. We classify

the sites into different categories such as marine boundary layer (4.5 ppb), terrestrial (25 ppb), mixed marine and terrestrial (15 ppb) and strong local influence (30 ppb). The uncertainties range from 4.5 to 75 ppb for global sites (Supplementary Table S1) and from 15 to 30 ppb for the Finnish sites (Table 1)."

*P6, L63:*

*With "natural prior from Saunois et al.", do you refer to wetland fluxes only or is it really all natural fluxes combined? It is a bit unclear because fire, termites, geological fluxes etc. are also natural $CH_4$ sources.*

It has now been specified:

> "In addition to the emissions from JSBACH-HIMMELI and LPX-Bern DYPTOP, the wetland prior (monthly averages from the 11 models used by Poulter et al., 2017) combined with the soil sink from Saunois et al. (2024) is used and referred here as the Global Carbon Project (GCP) prior."

*P6, L66:*

*In section 2.2, you have described well the atmospheric sinks that you have considered for the inversion. However, if I understand correctly, you do not include the $CH_4$ soil sink as negative prior emissions. Is that correct, and if so, could you explain why you have excluded them?*

Soil sink is calculated by the process-based models which estimates we use as prior. We have now specified in Section 2.2.4:

> "For natural prior emissions, we use estimates from two ecosystem models: Jena Scheme for Biosphere–Atmosphere Coupling in Hamburg with the HelsinkI Model of MEthane buiLd-up and emIssion for peatlands module (JSBACH-HIMMELI) (Raivonen et al., 2017;

Kleinen et al., 2020) and the Land surface Processes and eXchanges with the Dynamical Peatland Model Based on TOPMODEL (LPX-Bern DYPTOP) v1.4 (Lienert and Joos, 2018; Stocker et al., 2014; Spahni et al., 2011, 2013), which include $CH_4$ emissions from peatlands and mineral soils as well as the soil sink."

*P6, L175-L176:*

*It would be nice if you could add an example of the range of annual wetland emissions in numbers to make this more illustrative.*

Added as suggested:

"Studies based on process-based models have shown that estimates of $CH_4$ emissions from wetlands vary substantially and inhomogeneously (e.g., a global annual average was 119–203 Tg $yr^{-1}$ from 2010 to 2019; Saunois et al., 2024)…"

*P7, L198:*

*I think it would be good to also explain the reason for the different time periods (shown in Table 1) in this paragraph.*

We have now specified:

"The time periods covered by each inverse model run also differed depending on the priors used and the time periods they covered."

*P8, L211- P9, L29:*

*It would be helpful if you could give a little outlook, for what the different data types described in the subsections of 2.4 will be used for. Otherwise it is a bit incomprehensible, why you suddenly start writing about flux measurements, global freshwater emissions and fire inventories. The subsections are also so short that I would suggest combining them.*

We removed the numbering from the titles and added the following sentence:

> "To help us interpret the $CH_4$ emissions estimated by the inverse model, we use auxiliary $CH_4$ datasets introduced below."

*P11, L268:*

*With "Finland's annual total emission estimates" you are referring to the bottom-up/prior emissions I assume? If so, please add it for clarification.*

It is referring to both prior and posterior emissions. We now clarified the sentence:

> "The annual total emissions of Finland from the five CTE-$CH_4$ inverse model runs are shown in Fig. 3."

*P11, L269-L270:*

*Please re-write this sentence. I would suggest: "As discussed in section 3.1, the range of total prior emissions was large. The range of optimized emissions was smaller, especially between 2009 and 2020 with an average range of 0.57 Tg yr$^{-1}$, while the range of prior emissions was 0.69 Tg yr$^{-1}$ in the same period."*

The total emissions here are referencing the total $CH_4$ budget, not only anthropogenic. However, we modified the sentence in other respects:

> "The range of the total prior emissions was large, but the range of the optimised emissions was lower after 2009 and until 2020, with an average range of 0.57 Tg yr$^{-1}$ in 2009–2020, while the range of the prior emissions was 0.69 Tg yr$^{-1}$ in the same period."

*P12, L283-L286:*

*Please also re-write this sentence. I would suggest: "The order of the emission estimates was also maintained after optimization: the posterior emissions of Inv$_{JSBACH\_CAMSREG}$ were the highest and Inv$_{GCP\_EDGAR}$ the lowest. The three posterior emissions using LPX-Bern DYPTOP as prior lay between these two estimates, with the inversion using the varying uncertainty estimates (Inv$_{LPX\_EDGAR\_UNC}$) showing the lowest estimates of the three."*

Modified as suggested.

*P12, L288 – P14, L297:*

*I assume you forgot to put a references for Table 2 in this paragraph?*

Yes, we did. Thank you for the observation. It has now been added.

*P13, L306:*

*I would add "In 2021, there were only three posterior emission estimates" to make it clearer that there is a difference to 2020.*

Modified as suggested.

*P14, L307:*

*Could you clarify how the differences diverged between 2020 and 2021? And did you only use the EDGAR set-ups for comparison the two years?*

We tried to clarify this sections:

> "The three optimised total emissions were still higher than in 2019, but emissions from Inv$_{GCP\_EDGAR}$ were lower in 2021 than in 2020, while inversions with LPX-Bern DYPTOP were higher in 2021. The partitioning to natural or anthropogenic was also inconsistent across the three inversion estimates: Inv$_{GCP\_EDGAR}$ had both lower anthropogenic and natural emissions in 2021 than in 2020, while in Inv$_{LPX\_EDGAR\_UNC}$ it was the other way round. At the same time, the

anthropogenic emissions of $Inv_{LPX\_EDGAR}$ were higher and the natural emissions were lower in 2021 than in 2020. However, the differences between 2021 and 2020 were similar in magnitude to previous years."

*P17, L351-L352:*

*Are the small uncertainties in the winter months a result of the low natural emissions during those months? And could you clarify "In February and March, both the natural $CH_4$ emissions and the uncertainties were small regardless of the uncertainty estimates used" please? Do you mean "both the posterior natural $CH_4$ emissions and the uncertainties regardless of the prior uncertainty estimates used"?*

Yes, with the standard uncertainty estimate (80 % of the flux), the small uncertainty is due to low natural emissions in winter and with the uncertainty based on the ensemble of process-based models, it is due to the small range of process-based model estimates.

We modified the text to be as follows:

"From November to January, the uncertainty was smaller almost everywhere in $Inv_{LPX\_EDGAR\_UNC}$ than in $Inv_{LPX\_EDGAR}$. In February and March, both $Inv_{LPX\_EDGAR\_UNC}$ and $Inv_{LPX\_EDGAR}$ had low natural prior $CH_4$ emissions and small uncertainties."

*P19, L385 – P20, L424:*

*In section 3.4, you use the full names of the observation sites multiple times. However, the only reference that you provide for the sites is Figure 1, where only the abbreviations of the sites are given. As mentioned in my first general comment, a proper documentation of these observation sites would be very helpful and I would suggest to use the abbreviations of the sites in this section, so that they are easy to find on Figures 1 and 11.*

We modified the text as suggested. In addition, we added a table of the site-specific details (supplementary Table S1).

*P24, L499:*

*I assume the "CH$_4$ emissions" from Feng et al. refer to global CH$_4$ emission? I would suggest to add this information at the beginning of the sentence.*

Yes, it does. Added as suggested.

**Technical corrections**

We have modified the manuscript as suggested by the corrections below. Under some of the corrections we have added additional comments.

**E.g.** *P1, L18-L19, P2, L34-L35:*

*To make the paper more pleasant to read, I would suggest to avoid double brackets like "(15.15 ppb in 2020 and 17.97 ppb in 2021) (Lan et al., 2024)" and brackets inside brackets like "(CAMS-REG,(Kuenen et al., 2022))" or " (GAINS, Höglund-Isaksson et al. (2020))".*

*You could instead just put a comma: "(15.15 ppb in 2020 and 17.97 ppb in 2021, Lan et al., 2024)", "CAMS-REG, Kuenen et al., 2022)" and "(GAINS, Höglund-Isaksson et al., 2020)"*

We have now removed any double brackets and replaced them with citations including a semicolon, for example, "(CAMS-REG; Kuenen et al., 2022)".

*P2, L22:*

*I don't think "CH$_4$'s" is technically incorrect, but it looks a bit unusual. Maybe write "short atmospheric lifetime of CH$_4$" instead.*

*P2, L28-L29:*

*Please check the sentence structure. I would suggest to write "They are based on a bottom-up approach that starts at the sources and estimates how much GHGs is emitted by each source."*

*P2, L48:*

*"All countries" → "All **these** countries" makes it clearer*

*P4, L89:*

*"agrees **the** best with observations" → " agrees best with **the** observations."*

*P4, L95:*

*Why do you write "(Intergovernmental Panel on Climate Change, 2019)" instead of just "(IPCC, 2019)"?*

Changed to "IPCC"

*P4, L97:*

*"category, (LULUCF), are" → either write "category (LULUCF) are" or "category, LULUCF, are".*

Brackets removed.

*P7, L207:*

*"total $CH_4$ emission" → "total $CH_4$ emissions"*

*P8, Figure 1:*

*I'm a bit confused by the caption "black and old". The color of the circles looks rather gray and I assume you mean "bold"?*

*P12, P13 & P16, Fig. 5, 6 & 7:*

*I would suggest to also put "prior" and "posterior" in the legends as was the case in Fig. 3.*

*P12, L285:*

*$(Inv_{LPX\_EDGAR\_UNC}))$ → $(Inv_{LPX\_EDGAR\_UNC})$*

*P16, L324:*

*"posterior emission" → "posterior emissions"*

*P16, L125:*

*I'm not sure if "with the VERIFY ensemble" is correct. Do you mean "in the VERIFY ensemble"?*

*P16, Fig 7:*

*Maybe you could use a different color for the freshwater emissions? Despite the different line style, they are a bit hard to spot.*

***E.g.** P26, L552 and 555:*

*When talking about emissions, use "high" and "low" instead of "large" and "small".*

*### References*

*Thompson, R. L., Nisbet, E., Pisso, I., Stohl, A., Blake, D., Dlugokencky, E., Helmig, D., and White, J.: Variability in atmospheric methane from fossil fuel and microbial sources over the last three decades, Geophysical Research Letters, 45, 11–499, 2018*

*Rice, A. L., Butenhoff, C. L., Teama, D. G., Röger, F. H., Khalil, M. A. K., and Rasmussen, R. A.: Atmospheric methane isotopic record favors fossil sources flat in 1980s and 1990s with recent increase, Proceedings of the National Academy of Sciences, 113, 10 791–10 796, 2016.*